# CompoDistill: Attention Distillation for Compositional Reasoning in Multimodal LLMs

**Jiwan Kim**[1,✉], **Kibum Kim**[1,✉], **Sangwoo Seo**[1,✉], **Chanyoung Park**[1,✉*]
[1]KAIST

## Abstract

Recently, efficient Multimodal Large Language Models (MLLMs) have gained significant attention as a solution to their high computational complexity, making them more practical for real-world applications. In this regard, the knowledge distillation (KD) approach has emerged as a promising alternative, which transfers the rich visual and linguistic knowledge from a larger model (teacher) to a smaller model (student). However, we observe that existing KD methods struggle to effectively distill the teacher MLLM's rich *visual perception* abilities to the student, a challenge that has been largely overlooked in previous studies. Through a systematic analysis, we identify visual attention misalignment between student and teacher as the main cause of this issue. Based on this insight, we propose CompoDistill, a novel KD framework that explicitly aligns the student's visual attention with that of the teacher to enhance the student's visual perception abilities. Our extensive experiments show that CompoDistill significantly improves performance on compositional reasoning tasks that require visual perception abilities while maintaining strong performance on visual question answering tasks, as done in existing studies. Furthermore, CompoDistill demonstrates effectiveness with a more advanced backbone, highlighting its generalizability.

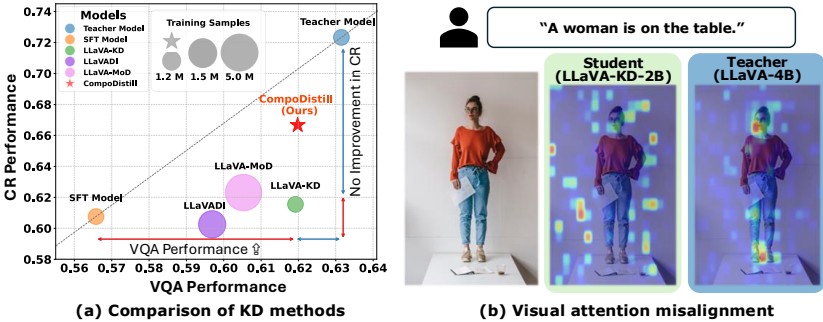

(a) Comparison of KD methods          (b) Visual attention misalignment

Figure 1: (a) Comparison of the teacher (LLaVA-4B), SFT (LLaVA-2B), and various 2B KD methods distilled from the same teacher. (b) *Visual attention misalignment* between the student (LLaVA-KD-2B) and teacher (LLaVA-4B), where the student attends to irrelevant image regions for the text query *A woman is on the table*, in contrast to the teacher. For more examples including our proposed method, CompoDistill, please refer to Appendix A.

# 1 Introduction

Multimodal Large Language Models (MLLMs) (Liu et al., 2023; Lin et al., 2024b) have demonstrated superior performance on various vision-language tasks over vision-language models (Radford et al., 2021; Zhai et al., 2023), marking a significant step forward in multimodal learning. Yet, their advancement has largely been driven by scaling laws, which has resulted in ever-larger and more computationally demanding architectures (OpenAI, 2023; Qwen et al., 2025). The substantial computational and memory costs associated with such massive models pose considerable challenges for practical deployment. This has, in turn, intensified interest in developing **efficient MLLMs** that can maintain strong multimodal capabilities while mitigating resource requirements.

---

*Corresponding author

To this end, **knowledge distillation (KD)** (Cai et al., 2024; Shu et al., 2024; Xu et al., 2024) has emerged as a promising approach, transferring rich knowledge from a larger *teacher* model to a smaller *student* model. These KD-based methods have demonstrated effectiveness in visual question answering (VQA) tasks (e.g., GQA (Hudson & Manning, 2019)), which require *visual recognition* ability, outperforming the standard supervised fine-tuning (SFT model[1] in Figure 1(a)).

Despite the progress of KD methods in visual recognition ability, a crucial question remains unexplored: *Is visual perception ability equally well distilled? Visual recognition* refers to tasks like object classification, where the goal is to identify objects in images based on their features. On the other hand, *Visual perception* involves more complex abilities, such as understanding relationships among objects and accurately capturing their attributes—capabilities essential for real-world multimodal applications[2]. To answer the question, in Figure 1(a), we compare the state-of-the-art KD methods (Shu et al., 2024; Cai et al., 2024; Xu et al., 2024) on VQA and compositional reasoning (CR) datasets, where VQA serves as a benchmark for evaluating visual recognition ability and CR is designed to assess visual perception ability[3]. We observe that, although existing KD methods show significant performance improvements on the VQA, their performance on the CR is on par with the SFT model, which is unexpected. Based on this observation, *we argue that existing KD methods struggle to effectively distill visual perception ability from the teacher model.*

Beyond a simple performance comparison, we investigate the failure of the existing KD methods in distilling visual perception ability and analyze the attention maps of both the student (i.e., LLaVA-KD-2B (Cai et al., 2024)) and the teacher (i.e., LLaVA-4B (Liu et al., 2023)) models. Specifically, by visualizing their focus areas for a given text, we aim to identify differences in attention, which we believe are closely related to perception ability. As shown in Figure 1(b), we observe that the student model struggles to focus on the relevant regions, while the teacher model captures them effectively, revealing a clear mismatch in the attention distributions between the teacher and the student models. The discrepancy in attention distributions, which we term *visual attention misalignment*, reveals that although KD methods aim to transfer knowledge from the teacher to the student through knowledge distillation, the student fails to inherit the teacher's powerful visual attention mechanism, thereby limiting the effective distillation of visual perception ability.

In this regard, we hypothesize that the limited performance improvement of KD methods in CR tasks arises from the visual attention misalignment between the teacher and student models. To empirically validate our hypothesis, we conduct controlled experiments examining the relationship between attention behavior and visual task performance in Section 3. Our results demonstrate that this misalignment is directly responsible for the unexpectedly low performance of the existing KD methods in CR tasks, and this study is the first to establish that addressing visual attention misalignment is the key to enhancing the student model's visual perception ability.

Motivated by these findings, we propose CompoDistill, a novel KD framework aimed at effectively distilling the teacher's rich visual perception abilities to the student model by addressing visual attention misalignment. We introduce the **Visual ATtention alignment (VAT)** module to explicit align the visual attention of the student model with that of the teacher model, using a simple yet effective group matching strategy, in which each student layer is matched with a group of teacher layers in a one-to-many manner, to handle the difference in the number of LLM layers between the teacher and student. However, the teacher's visual attention is optimized for its own vision-language space, making a simple transfer via the VAT module ineffective within the student's distinct and incompatible space. This mismatch creates a conflict between the student's feature space and the imposed attention mechanism, ultimately restricting the model's inherent perceptual abilities. To this end, we propose the **Teacher Adapter Fetch (TAF)** module to bridge this feature space gap and enable synergy with the VAT module. Building on this, we introduce a meticulously designed three-stage training strategy that leverages both modules to comprehensively distill visual perception.

Through extensive experiments on multiple CR datasets, we demonstrate that CompoDistill significantly outperforms existing KD methods, demonstrating the effectiveness of CompoDistill in enhancing the student model's visual perception ability. Meanwhile, CompoDistill maintains com-

---

[1]Sharing the same architecture and size as the student models in KD, this model is trained only with visual instruction tuning Liu et al. (2023) via supervised fine-tuning. We henceforth denote it as the SFT model.

[2]Regarding a detailed comparison of the differences between the two abilities, please refer to Appendix B.

[3]VQA datasets include GQA (Hudson & Manning, 2019), TextVQA (Singh et al., 2019), and MME (Fu et al., 2024), and CR datasets include SugarCrepe (Hsieh et al., 2023), SADE (Ma et al., 2023), BiVLC (Miranda et al., 2024), and Winoground (Thrush et al., 2022). Results in Figure 1(a) correspond to the average performance across the datasets.

petitive performance on VQA datasets, preserving its visual recognition abilities. Furthermore, we demonstrate the effectiveness of CompoDistill with a more advanced backbone, highlighting its generalizability.

We summarize our contributions as follows: (1) We identify, for the first time, that existing KD methods in MLLMs fail to distill visual perception ability from the teacher and provide a detailed attention-based analysis, offering insights on how to enhance this ability. (2) We propose CompoDistill, a novel KD framework that transfers both visual recognition and perception abilities through the Visual ATtention alignment and the Teacher Adapter Fetch Module. (3) We achieve significant improvements in CR tasks by effectively distilling the visual perception ability, while maintaining competitive performance in VQA tasks by preserving the visual recognition ability.

## 2 PRELIMINARY

**Multimodal Large Language Models (MLLMs).** Following the LLaVA (Liu et al., 2023) design, MLLMs consist of three main components: a pre-trained LLM $LM_\theta(\cdot)$, a vision encoder $V_\phi(\cdot)$, and an adapter $P_\psi(\cdot)$. Given a text query $Q$ and an image $I \in \mathbb{R}^{H \times W \times 3}$, where $H$ and $W$ denote the image height and width, the vision encoder extracts patch-level features $\mathbf{z}_p = V_\phi(I) \in \mathbb{R}^{N_v \times d_p}$, with $N_v$ being the number of visual patches and $d_p$ the hidden dimension of $V_\phi$. The adapter projects them into the language space as $\mathbf{x}_v = P_\psi(\mathbf{z}_p) \in \mathbb{R}^{N_v \times d}$, where $d$ is the hidden dimension of $LM_\theta$. Hereafter, we refer to $\mathbf{x}_v$ as the *visual tokens*. Meanwhile, the query $Q$ is tokenized into embeddings $\mathbf{x}_t \in \mathbb{R}^{N_t \times d}$, where $N_t$ is the number of text tokens. The combined sequence $[\mathbf{x}_v, \mathbf{x}_t] \in \mathbb{R}^{(N_v + N_t) \times d}$ is processed by the $LM_\theta$ to compute the probability of the target answer as:

$$p(\mathbf{y}_{1:K}) = \prod_{i=1}^{K} p(\mathbf{y}_i \mid \mathbf{x}_v, \mathbf{x}_t, \mathbf{y}_{<i}), \tag{1}$$

where $\mathbf{y}_{<i}$ is the sequence of the answer tokens up to the $i$-th token and $K$ is the answer length. During training, the cross-entropy loss derived from Equation 1 is minimized, denoted as $\mathcal{L}_{LM}$.

## 3 WHY IS THE VISUAL PERCEPTION ABILITY NOT DISTILLED PROPERLY?

In Section 1, we noted that existing KD methods, despite improvements in terms of VQA, show unexpected results on compositional reasoning (CR) tasks, which we attribute to ***visual attention misalignment***. This section provides an in-depth analysis to elucidate the underlying causes of this issue. Our analysis first identifies the key factor for distilling visual ability through an attention-based analysis (Section 3.1), then demonstrates this factor's direct impact on performance (Section 3.2). Finally, we validate that alleviating the visual attention misalignment associated with this key factor facilitates a more effective transfer of visual perception (Section 3.3).

### 3.1 IDENTIFYING THE KEY FACTOR: TEACHER-STUDENT ATTENTION SIMILARITY OVER VISUAL TOKENS IN THE VISUAL UNDERSTANDING LAYER

In this section, we aim to identify a critical factor that can serve as a criterion for successfully distilling visual abilities (i.e., visual recognition and perception). To achieve this, we examine the layer-wise functions of the LLM within MLLMs, building on insights from prior works (Chen et al., 2025a; Yoon et al., 2025). This is followed by an attention-based analysis to pinpoint the key factor.

**Layer-wise Functionality in MLLMs.** Following previous studies (Chen et al., 2025a; Yoon et al., 2025), we adopt a layer-wise view of MLLM processing: *Early layers* align heterogeneous modalities with the LLM's text space, *Intermediate layers* integrate these signals for fine-grained semantic understanding, and *Later layers* generate the final response. Our focus is on the intermediate layers[4], which we term the **visual understanding layers**, since these layers play a critical role in forming foundational visual reasoning and comprehension (Chen et al., 2025b), which are closely tied to the visual abilities (recognition and perception) central to our study.

**Attention Analysis for VQA and CR Tasks.** Building on our focus on the visual understanding layers, we design an experiment to investigate if teacher-student attention alignment can explain the difference in the degree of performance improvement between VQA and CR tasks, as observed in Figure 1(a). To this end, we compare the attention distribution of the teacher model (i.e., LLaVA-4B (Liu et al., 2023) with that of two distinct models: a student model distilled directly from the

---

[4]Following (Neo et al., 2025), intermediate layers refer to the 30–70% range of the total LLM layers.

teacher (i.e., LLaVA-KD-2B (Cai et al., 2024)) and an SFT model (i.e., LLaVA-2B) that is architecturally identical to the student model but trained without distillation.

As shown in Figure 2(a), we measure the attention similarity to quantify the alignment between a given model (student or SFT) and the teacher. This is computed as the cosine similarity of their attention distributions, where the answer token $\mathbf{y}_i$ is the *query (Q)* and the visual tokens $(\mathbf{x_v})$ or text tokens $(\mathbf{x_t})$ are the *key (K)*. Our primary analysis centers on the attention distributions over visual tokens during answer generation, as these are critical for the visual abilities being studied. For a more comprehensive comparison, we also investigate attention distribution over text tokens. We analyze the attention similarity of the student and SFT models to the teacher.

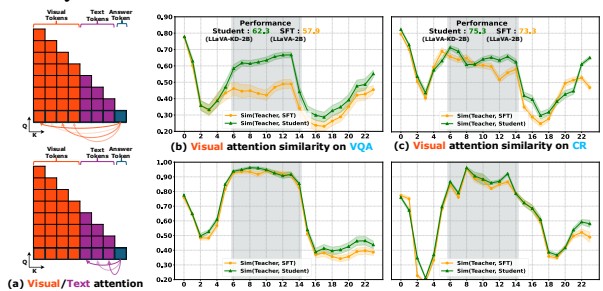

Specifically, we investigate how these similarities in attention behavior are related with each model's performance in VQA and CR task, which brings us closer to identifying the key factor. To guide this investigation, we formulate two key research questions:

***Q1) Where do the performance gains of KD on VQA (visual recognition) tasks originate?*** For VQA, where the student model demonstrates clear performance improvements, Figure 2(b) shows that, within the *visual understanding layers*, the teacher-student attention similarity over visual tokens is significantly higher than that between the teacher-SFT. However, no such improvement is observed for text tokens (Figure 2(d)). This finding brings us to identify the clue of the key factor: *if the teacher-student visual attention similarity in the visual understanding layers is high, distillation is effectively achieved, resulting in performance improvement of the student* .

Figure 2: (a) Attention of the answer token over visual tokens and text tokens during its generation. The transparency indicates the degree of attention the answer token gives to these tokens. (b-e) Layer-wise Teacher-Student and Teacher-SFT attention similarities over visual tokens ((b), (c)) and text tokens ((d), (e)). GQA is used for VQA ((b), (d)), and SugarCrepe is used for CR ((c), (e)). Results on other datasets are shown in Appendix C.

***Q2) Why does KD fail to deliver comparable improvements on CR (visual perception) tasks?*** We extend our analysis to CR tasks, where the student model performs on par with the SFT model. As shown in Figure 2(c) and (e), we observe that the teacher-student attention similarity is comparable to the teacher-SFT attention similarity in the *visual understanding layers*, even over the visual tokens, which contrasts with our observation on VQA. That is, in the *visual understanding layers*, the student model shows no better alignment with the teacher model in terms of the attention over visual tokens than the SFT model, despite such alignment being crucial for success in the downstream performance as shown in the case of VQA. Drawing on the findings from ***Q1*** and ***Q2***, we argue that the inability of existing KD methods to effectively distill visual perception ability stems from the moderate level of teacher–student attention similarity in the *visual understanding layers*.

## 3.2 RELATIONSHIP BETWEEN TEACHER-STUDENT ATTENTION SIMILARITY AND DOWNSTREAM PERFORMANCE

While the teacher-student attention similarity over visual tokens appears to be a key factor in determining whether visual abilities have been distilled, its direct relationship with downstream performance remains unclear. In other words, it is uncertain whether the higher teacher–student attention similarity (green line), relative to the teacher–SFT attention similarity (yellow line) in the visual understanding layers (Figure 2(b)), explains the superior performance of the student model on the VQA dataset. To this end, we quantify the relationship between the teacher-student *visual attention similarity* and the *answer*

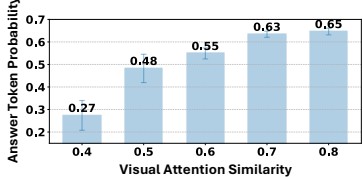

Figure 3: Relationship between attention similarity and performance on the GQA (VQA) dataset.

*token probability* given in Equation 1 on the VQA dataset (Figure 3), where visual recognition ability has been successfully distilled from the teacher model. More precisely, we randomly sampled 5,000 instances from the GQA test set, grouped them according to the student–teacher similarity over visual tokens during the answer generation step, and then measured the average probability assigned to the ground-truth answers. The results reveal a clear positive trend: higher attention similarity is

consistently associated with higher answer probabilities, providing direct evidence that aligning the student's attention with the teacher's attention is a key factor driving better performance on VQA.

### 3.3 A SIMPLE SOLUTION: REPLACING STUDENT'S VISUAL ATTENTION WITH TEACHER'S

**Is Teacher Attention Really Beneficial?** A natural question arises: *Does increasing the teacher-student attention similarity over visual tokens —a key factor for VQA performance—also enhance performance on CR tasks?* To test this, we perform a direct intervention during inference on CR. Before generating the answer, we substitute the student's original attention over visual tokens with the average of the teacher's and student's attention, thereby increasing the student's attention similarity to the teacher[5].(Figure 4).This yields modest but consistent performance gains in the SugarCrepe dataset (Swap, Replace and Add are three types of CR sub-tasks in the dataset). Although small, these improve-

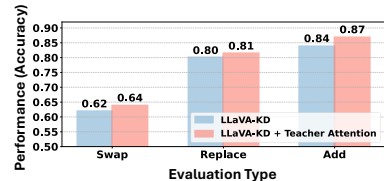

Figure 4: Change in performance the student attention is substituted with the teacher attention on the SugarCrepe (CR) dataset.

ments confirm that the student benefits from incorporating the teacher's attention, reinforcing our hypothesis that attention alignment plays a crucial role in effective knowledge transfer.

In summary, our analysis identifies *visual attention misalignment* as the key barrier to effectively distilling the teacher's perception abilities to the student, highlighting the crucial need to accurately transfer the teacher's visual attention.

## 4 METHODOLOGY: COMPODISTILL

Motivated by the findings in Section 3, our framework, called CompoDistill, is designed to effectively transfer the teacher's visual perception abilities by addressing the visual attention misalignment. We first introduce our two core components: the **Visual ATtention alignment (VAT)** module (Section 4.1), which aligns the student's visual attention mechanism with that of the teacher, and the **Teacher Adapter Fetch (TAF)** module (Section 4.2), which ensures the student processes visual space consistently with the teacher. These modules are then integrated into a meticulously designed three-stage training strategy (Section 4.3). The overall framework is shown in Figure 5.

### 4.1 VISUAL ATTENTION ALIGNMENT (VAT) MODULE

**Attention Distillation Loss.** To transfer the teacher's attention mechanism to the student, we utilize the attention matrix of the Transformer (Vaswani et al., 2017) within the LLM layer, which represents the importance of each token relative to other tokens. The attention matrix for the input tokens (i.e., $\mathbf{x}_v$ and $\mathbf{x}_t$) is computed as: $A = \text{softmax}\left(\mathbf{Q}\mathbf{K}^\top/\sqrt{d}\right) \in \mathbb{R}^{(N_v+N_t)\times(N_v+N_t)}$, where $\mathbf{Q} \in \mathbb{R}^{(N_v+N_t)\times d}$ and $\mathbf{K} \in \mathbb{R}^{(N_v+N_t)\times d}$ are the query and key matrices derived from the input tokens, respectively. From the overall attention matrix $A$, as discussed in Section 3, our goal is to distill the *attention over visual tokens* from the teacher to the student in the visual understanding layers, leading us to extract a sub-matrix $\tilde{A} \in A$ that focuses on visual attention. Specifically, for each visual understanding layer $l$, this sub-matrix includes only the columns (keys) of $A_l$ corresponding to the visual tokens, resulting in $\tilde{A}_l = A_l[:,:N_v] \in \mathbb{R}^{(N_v+N_t)\times N_v}$. Based on the visual attention-related sub-matrices of the teacher ($\tilde{A}_l^t$) and student ($\tilde{A}_l^s$), we compute the cosine distance between them for the attention distillation loss as $1 - \text{sim}\left(\tilde{A}_{l_t}^t, \tilde{A}_{l_s}^s\right)$, where $l_t$ and $l_s$ denote the index of the visual understanding layers of the teacher and student, respectively.

**Group Layer Matching.** However, since the teacher has more layers than the student, directly matching a student layer index $l_s$ with a teacher layer index $l_t$ is challenging. A naive solution is to map each $l_s$ to one $l_t$ by uniformly sampling teacher layers according to the ratio of their depths. For example, if the student has 5 layers and the teacher has 10 layers, then teacher layers $\{1, 3, 5, 7, 9\}$ are selected and aligned with the 5 student layers. However, this approach is suboptimal, as it overlooks the richer perception abilities distributed across the teacher's layers and does

---

[5]We initially attempted to completely replace the student's attention with that of the teacher. However, this rather led to a slight performance degradation compared with vanilla LLaVA-KD, which we attribute to a mismatch between the student's feature space and the teacher's attention.

not ensure accurate alignment[6]. Instead, we propose a simple yet effective **Group Layer Matching** strategy, where each $l_s$ is aligned with a group of $\{l_t\}$ in a one-to-many manner, enabling the student to capture broader teacher knowledge while roughly preserving the layer order.

Formally, let the sequence of student's visual understanding layer indices be $L_S = \{l_s^1, \ldots, l_s^j, \ldots, l_s^k\}$ and that of the teacher be $L_T = \{l_t^1, \ldots, l_t^j, \ldots, l_t^m\}$, where $j$ is the $j$-th layer in the visual understanding layers, and $k$ and $m$ denote the total number of visual understanding layers for the student and teacher, respectively, with $k < m$. For each student layer $l_s^j$, we define a corresponding group of teacher layers, $G_j$, consisting of $n$[7] consecutive teacher layers, formed using a sliding window approach. For example, the group of teachers $G_1$ assigned for the first student layer $l_s^1$ could be $\{l_t^1, l_t^2, \ldots, l_t^n\}$, while the group $G_2$ for the second student layer $l_s^2$ would be $\{i_t^2, i_t^3, \ldots, i_t^{n+1}\}$, and so on. So, the total number of groups is the same as the number of student's target layers, $k$.

To distill the visual attention of a teacher group $G_j$ into student layer $l_s^j$, we formulate the objective by minimizing the cosine distance between the attention matrix of student layer, $\tilde{A}_{l_s^j}^s$, and the averaged attention matrices of its corresponding teacher group, $\bar{A}_j^t$, as follows:

$$\mathcal{L}_{ADL} = 1 - \frac{1}{k} \sum_{j=1}^{k} \mathrm{sim}\left(\bar{A}_j^t, \tilde{A}_{l_s^j}^s\right), \quad \bar{A}_j^t = \frac{1}{n} \sum_{l \in G_j} \tilde{A}_l^t. \quad (2)$$

Furthermore, beyond the naive one-to-one matching approach, our proposed group matching strategy is superior to the more advanced adaptive matching method (Lee et al., 2025b) in terms of effectiveness, as demonstrated in Section 5.3.

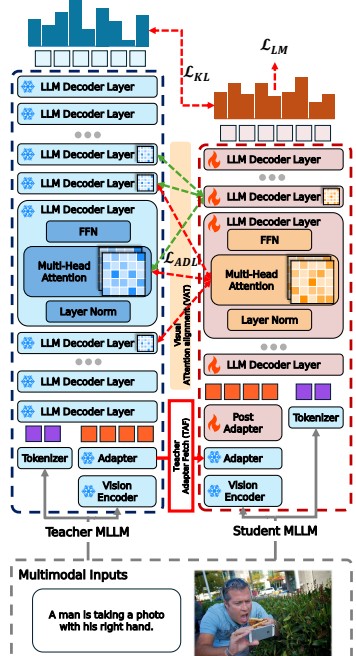

## 4.2 TEACHER ADAPTER FETCH (TAF) MODULE

The adapter, responsible for projecting the visual space into the language space of the LLM, is crucial for generating the visual tokens with which the attention mechanism processes. In this regard, given that the teacher's attention mechanism transferred via VAT is tightly coupled with the output of its own adapter, a vision-language space mismatch occurs when this attention is imposed on a student with a different, incompatible space, hindering effective knowledge transfer. To address this, we introduce the **Teacher Adapter Fetch (TAF)** module, which directly leverages the teacher's frozen, pretrained adapter ($P_{\psi_t}^t$) and adds a lightweight trainable MLP ($P_{\psi_s}^s$) only for dimensional alignment. This ensures the student processes the visual input through the same lens as the teacher, making the attention transfer effective. Formally, the student's visual token is expressed as follows:

Figure 5: Overview of CompoDistill. It consists of the VAT module and the TAF module.

$$\mathbf{x}_v = P_{\psi_s}^s \left( P_{\psi_t}^t(\mathbf{z}_p) \right) \in \mathbb{R}^{N_v \times d^s}, \quad (3)$$

where $d_s$ is the hidden dimension of the student LLM. This approach allows for the student to bridge the gap between the teacher-imposed attention mechanism and its own visual feature space.

**Discussion.** It is important to note that attention distillation itself is not a novel concept, as similar approaches have been explored in various domains (Wang et al., 2020; Sajedi et al., 2023; Zhou et al., 2025), including diffusion models, dataset distillation, and language models. However, its application to MLLMs is particularly challenging due to mismatched visual feature spaces between teacher and student. To this end, we introduce the TAF module as a simple yet effective solution that makes attention distillation practical in this setting. Lastly, our work is not intended to propose a fundamentally new distillation method; rather, its primary contribution lies in identifying a critical factor that can serve as a criterion for successfully distilling visual abilities in MLLMs, i.e., visual attention misalignment, as explained in Section 3.

---

[6]In Table 3, we show that this approach is less effective than the proposed Group Layer Matching strategy.

[7]To ensure the use of all teacher layers $L_T$, we define $n$ in closed form as $m - k + 1$.

## 4.3 THREE-STAGE DISTILLATION FRAMEWORK

Our knowledge distillation framework consists of three stages, built upon two foundational training objectives. The first objective is language modeling autoregressive loss $\mathcal{L}_{LM}$, as defined in Section 2. Moreover, following previous studies (Shu et al., 2024; Cai et al., 2024), we employ a Kullback-Leibler (KL) divergence loss ($\mathcal{L}_{KL}$) to align the student's predictive distribution ($p_s$) with the teacher's ($p_t$), encouraging the student to mimic the teacher's final output at the logit level:

$$\mathcal{L}_{KL} = -\frac{1}{K} \sum_{k=1}^{K} \sum_{n=1}^{N} p_t(\mathbf{y}_n|\mathbf{x}_v, \mathbf{x}_t, \mathbf{y}_{<k}) \log \frac{p_t(\mathbf{y}_n|\mathbf{x}_v, \mathbf{x}_t, \mathbf{y}_{<k})}{p_s(\mathbf{y}_n|\mathbf{x}_v, \mathbf{x}_t, \mathbf{y}_{<k})}, \tag{4}$$

where $\mathbf{y}_n$ is the $n$-th vocabulary token and $N$ is the total vocabulary size.

**Stage 1: Distilled Pre-Training (DPT).** The first stage aims to align the visual feature space with the language space. To this end, instead of initializing an adapter from scratch, we construct it using our **Teacher Adapter Fetch** module, which reuses the teacher's pretrained adapter, which is kept frozen during training. While the vision encoder and LLM remain frozen, the student adapter $P_{\psi_s}^s$ is optimized with the language modeling loss ($\mathcal{L}_{LM}$) and the KL-divergence loss ($\mathcal{L}_{KL}$).

**Stage 2: Distilled Fine-Tuning (DFT).** In the second stage, we aim to enhance the student's visual perception by aligning its visual attention mechanism over visual tokens with that of the teacher via the **Visual ATtention Alignment** module. During this stage, both the student LLM and the student adapter $P_{\psi_s}^s$ are fine-tuned. The overall objective is defined as $\mathcal{L}_{LM} + \mathcal{L}_{KL} + \mathcal{L}_{ADL}$.

**Stage 3: Supervised Fine-Tuning (SFT).** Finally, motivated by prior work (Lee et al., 2025b), we fine-tune only the student using the autoregressive language modeling loss ($\mathcal{L}_{LM}$), with both the student LLM and the student adapter $P_{\psi_s}^s$ trained in the same manner as in Stage 2. This final stage consolidates the rich knowledge transferred from the teacher during Stages 1 and 2 into the student's own parameters and further strengths its instruction-following capability.

## 5 EXPERIMENTS

### 5.1 EXPERIMENTAL SETTINGS.

**Evaluation Benchmarks.** For evaluation, we use two categories of vision-language tasks. ***General VQA***: We evaluate visual recognition abilities using well-established benchmarks including VQAv2 (Goyal et al., 2017), GQA (Hudson & Manning, 2019), and VizWiz (Bigham et al., 2010) for question answering, TextVQA (Singh et al., 2019) for scene text comprehension, and MME (Fu et al., 2024) for comprehensive multimodal evaluation. ***Compositional Reasoning***: To evaluate visual perception abilities, we use several challenging benchmarks: SugarCrepe (Hsieh et al., 2023), SADE (Ma et al., 2023), BiVLC (Miranda et al., 2024), and Winoground (Thrush et al., 2022). We use accuracy as the evaluation metric. Refer to Appendix D for more details regarding the baselines.

**Implementation Details.** Both the student and teacher employ SigLIP (Zhai et al., 2023) vision encoder and the Qwen 1.5 (Yang et al., 2024) LLM series, with the student having 1.8B parameters and the teacher having 4B parameters. Regarding the details of the training datasets and hyperparameter settings, please refer to Appendix E.

### 5.2 QUANTITATIVE RESULTS ON VQA AND CR TASKS

In Table 1, we compare CompoDistill with a diverse range of MLLMs of different sizes on VQA and CR tasks, which evaluate visual recognition and perception abilities, respectively. We have the following key observations: **1)** Existing KD methods[8] outperform the SFT model (LLaVA-2B) on VQA (visual recognition) tasks, but not on CR (visual perception) tasks, where their performance remains comparable to that of standard 2B models. This indicates that KD methods struggle to distill visual perceptual abilities. **2)** CompoDistill overcomes this limitation by significantly improving CR performance to a level competitive with 4B models—an achievement not attained by previous KD methods—while simultaneously maintaining strong, state-of-the-art performance on VQA. This demonstrates the ability of CompoDistill to enhance visual perception while maintaining visual recognition. **3)** CompoDistill achieves these results with high data efficiency, relying on just

---

[8]To ensure fair comparisons, all KD models were distilled from the same teacher model (i.e., LLaVA-4B) and share the same LLM backbone (i.e., Qwen 1.5).

Table 1: Comparison CompoDistill with KD-based methods and other MLLMs on VQA and CR. # Samples : Training data samples. [†]:Reproduced results using the official source code. The best and second-based models are marked in bold and underlined, respectively for models sizes under 2B.

| Size | Method | LLM | # Samples | Visual Question Answering | | | | | | Compositional Reasoning | | | | |
|---|---|---|---|---|---|---|---|---|---|---|---|---|---|---|
| | | | | VQAv2 | VizWiz | GQA | TextVQA | MME | Avg | Sugarcrepe | SADE | BiVLC | Winoground | Avg |
| ~7B | LLaVA-7B[†] | Qwen1.5-7B | 1.2 M | 79.8 | 53.1 | 62.3 | 60.1 | 68.6 | 64.8 | 87.3 | 78.5 | 65.7 | 58.4 | 72.5 |
| | LLaVA-7B[†] | Qwen2.5-7B | 1.2 M | 81.4 | 50.6 | 63.8 | 64.6 | 77.5 | 67.6 | 93.1 | 82.9 | 68.3 | 68.2 | 78.1 |
| | CogVLM-7B | Vicuna-7B | 1500 M | 82.3 | - | 64.9 | 78.2 | 71.8 | - | 81.8 | 63.2 | 64.5 | 57.2 | 66.7 |
| | Qwen2.5-VL-7B | Qwen2-7B | 1500 M | 82.8 | 61.7 | 63.9 | 73.9 | 83.9 | 73.2 | 88.5 | 77.4 | 68.4 | 75.3 | 77.4 |
| | Deepseek-VL-7B | DLLM-7B | 2000 M | - | 49.9 | 61.3 | 64.7 | 73.4 | - | 86.2 | 78.8 | 66.2 | 61.7 | 73.2 |
| ~4B | LLaVA-4B[†] (*Teacher*) | Qwen1.5-4B | 1.2 M | 79.1 | 48.0 | 62.1 | 56.7 | 67.2 | 62.6 | 83.0 | 75.5 | 64.8 | 57.8 | 70.3 |
| | Qwen2.5-VL-3B | Qwen2-3B | 1.2 M | 80.4 | 54.1 | 60.9 | 68.5 | 72.8 | 67.3 | 57.1 | 65.9 | 67.0 | 63.5 | 63.4 |
| | Imp-3B | Phi2-2.7B | 1.5 M | 81.2 | 54.1 | 63.5 | 59.8 | 72.3 | 66.2 | 78.1 | 61.1 | 59.5 | 38.3 | 59.3 |
| | Bunny-3B | Phi2-2.7B | 2.6 M | 79.8 | 43.8 | 62.5 | 56.7 | 74.4 | 63.4 | 75.8 | 67.2 | 59.7 | 53.1 | 64.0 |
| | MoE-LLaVA-3B | Phi2-2.7B | 2.6 M | 79.9 | 43.7 | 62.6 | 57.0 | 71.6 | 63.0 | 80.5 | 70.4 | 64.4 | 54.1 | 67.3 |
| | MobileVLM-3B | MobileLLaMA-2.7B | 1.3 M | - | - | 61.1 | 57.5 | 72.0 | - | 67.9 | 72.1 | 57.2 | 43.9 | 60.3 |
| | MiniCPM-V | MiniCPM-2.4B | 570 M | - | 60.2 | 52.1 | 73.2 | 70.5 | - | 90.0 | 70.6 | 65.3 | 55.3 | 70.3 |
| ~2B | MiniGemini-2B | Gemma-2B | 2.7 M | - | 41.5 | 60.7 | 56.2 | 67.0 | - | 67.0 | 62.0 | 58.0 | 50.5 | 59.4 |
| | Deepseek-VL-1.3B | DLLM-1.3B | 2000 M | - | 36.8 | 59.3 | **58.4** | 65.3 | - | 36.6 | 38.6 | 39.9 | 60.8 | 44.0 |
| | MobileVLM-1.7B | MobileLLaMA-1.4B | 1.2 M | - | - | 59.3 | 52.1 | 65.1 | - | 69.8 | 64.0 | 53.8 | 43.4 | 57.7 |
| | Imp-2B | Qwen1.5-1.8B | 1.5 M | **79.2** | 39.2 | 61.9 | 54.5 | 65.2 | 60.0 | 71.0 | 59.6 | 58.5 | 51.1 | 60.1 |
| | Bunny-2B | Qwen1.5-1.8B | 2.6 M | 76.6 | 34.2 | 59.6 | 53.2 | 65.0 | 57.7 | 68.6 | 66.5 | 57.8 | 48.5 | 60.3 |
| | MoE-LLaVA-2B | Qwen1.5-1.8B | 2.2 M | 76.2 | 32.6 | 61.5 | 48.0 | 64.6 | 56.6 | 80.8 | 62.5 | 62.0 | 50.6 | 63.9 |
| | LLaVA-2B (*SFT model*) | Qwen1.5-1.8B | 1.2 M | 73.6 | 31.2 | 57.9 | 50.6 | 61.2 | 54.9 | 73.3 | 63.9 | 58.5 | 47.2 | 60.7 |
| | LLaVA-KD-2B | Qwen1.5-1.8B | 1.2 M | 78.8 | **44.8** | **62.3** | 53.4 | 68.9 | 61.6 | 75.3 | 63.6 | 57.9 | 49.3 | 61.5 |
| | LLaVADI-2B[†] | Qwen1.5-1.8B | 1.2 M | 75.8 | 35.2 | 58.7 | 49.4 | 63.8 | 56.6 | 76.9 | 61.5 | 54.1 | 48.4 | 60.2 |
| | LLaVA-MoD-2B[†] | Qwen1.5-1.8B | 5.0 M | 76.3 | 43.0 | 58.8 | 52.7 | 63.4 | 58.9 | 76.9 | 63.8 | 60.3 | 49.4 | 62.6 |
| | CompoDistill-2B | Qwen1.5-1.8B | 1.2 M | 78.8 | 42.5 | 62.2 | 56.4 | **70.1** | 61.9 | **82.9** | **69.4** | **63.3** | **51.2** | **66.7** |

1.2M training samples. This is a sharp contrast to other models requiring much larger datasets (e.g., LLaVA-MoD with 5M, MiniCPM-V with 570M), proving the effectiveness of CompoDistill toward a truly efficient MLLM.

Table 2: Ablation for VAT and TAF modules.

| Row | VAT module | TAF module | Visual Question Answering | | | | Compositional Reasoning | | | | |
|---|---|---|---|---|---|---|---|---|---|---|---|
| | | | GQA | TextVQA | MME | Avg | Sugarcrepe | SADE | BiVLC | Winoground | Avg |
| (a) | ✗ | ✗ | 57.8 | 50.1 | 62.5 | 56.8 | 76.3 | 64.9 | 61.7 | 48.7 | 62.9 |
| (b) | ✓ | ✗ | 54.3 | 54.6 | 64.8 | 57.9 | 81.0 | 66.4 | 62.1 | 50.5 | 65.0 |
| (c) | ✗ | ✓ | 60.8 | 56.5 | 66.5 | 61.3 | 78.0 | 67.0 | 62.4 | 48.1 | 63.8 |
| (d) | ✓ | ✓ | 62.2 | 56.4 | 70.1 | 62.9 | 82.9 | 69.4 | 63.3 | 51.1 | 66.7 |

## 5.3 ABLATION STUDIES

**Effect of Core Components.** In Table 2, we conduct ablation studies to understand the effect of each component (i.e., VAT and TAF modules). Note that the variant without either component (row (a)) follows the proposed three-stage framework (Section 4.3), using only the language modeling loss (Section 2) and logit-based KD loss (Equation 4). *Effect of VAT module*: We observe that the VAT module significantly improves the performance especially on CR tasks (row (a) vs. (b) and row (c) vs. (d)), demonstrating the effectiveness of enhancing visual perception abilities. This confirms that explicit visual attention alignment is crucial for distilling visual perception ability, as discussed in Section 3. *Effect of TAF module*: We observe that equipping the TAF module improves the performance on VQA and CR tasks (row (a) vs. (c) and row (b) vs. (d)), indicating that bridging the student's feature space with the imposed attention mechanism is crucial for effective knowledge transfer. Finally, the fully-fledged model (row (d)) achieves the best performance on both VQA and CR tasks, demonstrating the benefit of the VAT and TAF modules.

Table 3: Detailed ablation for VAT module. The blue line performs the same as CompoDistill.

(a) **Attention loss type**

| Attention Loss | VQA Avg | CR Avg |
|---|---|---|
| ✗ | 61.3 | 63.8 |
| MSE | 60.3 | 65.2 |
| KL. Div. | 60.7 | 65.5 |
| Cos. Sim. | 62.9 | 66.7 |

(b) **Target layers**

| Target Layers | VQA Avg | CR Avg |
|---|---|---|
| Early (~ 30%) | 61.2 | 63.7 |
| Later (70% ~) | 61.7 | 64.6 |
| All | 62.4 | 66.6 |
| Intermediate | 62.9 | 66.7 |

(c) **Layer matching strategy**

| Match strategy | VQA Avg | CR Avg |
|---|---|---|
| Simple | 61.5 | 65.6 |
| Adaptive | 62.0 | 65.7 |
| Group | 62.9 | 66.7 |

**Fine-grained Analysis on VAT module.** We perform fine-grained ablation studies to study the impact of specific design choices—namely, the attention loss type, the target matching layer, and the layer matching strategy—within the VAT module. **1)** We first analyze the impact of the ***attention loss function*** (Table 3a). While any form of attention loss improves CR over the baseline, using Cosine Similarity (Cos. Sim.) significantly outperforms both MSE and KL Divergence. This result suggests that it is more crucial for the student to learn the relative importance among visual patches

rather than forcing an exact match of their absolute attention scores. **2)** Next, we investigate **which layers to target for distillation** (Table 3b). Performing distillation on the intermediate layers (30-70%) yields the highest performance. This finding confirms our analysis that *visual understanding layers* are crucial for visual-semantic integration and highlights the effectiveness of distilling specifically from these layers, a strategy consistent with prior research (Kaduri et al., 2024). **3)** Lastly, we evaluate different **layer matching strategies** (Table 3c). Our proposed Group matching achieves the best performance compared to both Simple matching (which uniformly samples teacher layers) and Adaptive matching (which finds optimal pairs based on layer distance). This suggests that grouping layers provides a more stable and effective signal for transferring the teacher's complex attention behaviors, especially when student and teacher architectures differ in depth.[9]

**Further Benefit on Other Complex Task.** Beyond compositional reasoning, we explore a further benefit of enhancing visual perception for complex tasks. Specifically, we expect that CompoDistill can help mitigate the relational hallucinations, thanks to its ability to accurately understand object relationships, as discussed in Section 1. To test this, we evaluate CompoDistill on the R-Bench (Wu et al., 2024) and Reefknot (Zheng et al., 2025) benchmarks, both of which focus on evaluating relational hallucinations.

Table 4: Comparison for relational hallucination using F1 score.

| Model | R-Bench ↑ | Reefknot ↑ |
|---|---|---|
| Teacher (LLaVA-4B) | 79.1 | 67.9 |
| Student (LLaVA-2B) | 74.3 | 61.3 |
| LLaVA-KD-2B | 76.5 | 60.3 |
| LLaVA-MoD-2B | 76.2 | 63.4 |
| CompoDistill-2B | 78.6 | 66.7 |

As shown in Table 4, CompoDistill significantly outperforms other KD methods and achieves performance nearly on par with the teacher. The results highlight that enhancing the visual perception ability can be beneficial to not only compositional reasoning tasks but also other complex tasks that require precise understanding of relationships among objects.

## 6 SCALABILITY EXPERIMENTS

**Richer Data Improves Distillation Performance.** We analyze the effect of data scale in Table 5. Performance improves significantly when moving from SFT to distillation, and increases further when the training dataset size is doubled[10]. This highlights that both the quality and quantity of training data are crucial for effective knowledge transfer, with CR showing particular sensitivity to data scaling.

Table 5: Data Scaling Experiment.

| Student | # Sample | VQA Avg | CR Avg |
|---|---|---|---|
| Qwen1.5-1.8B (SFT) | 1.2 M | 56.6 | 60.7 |
| Qwen1.5-1.8B | 1.2 M | 62.9 | 66.7 |
| Qwen1.5-1.8B | 2.4 M | 63.3 | 69.9 |

**Larger Teachers Produce Stronger Students.** Next, we examine the influence of teacher model size in Table 6. Our experiments show that students consistently benefit from larger teachers, regardless of the student's own size. For instance, a 1.8B student distilled from a 7B teacher outperforms one distilled from a 4B teacher. This demonstrates that a higher-capacity teacher is crucial for transferring stronger reasoning abilities, providing a more effective source of knowledge for the student.

Table 6: Experiments with different size of teacher/student.

| Student | Teacher | VQA Avg | CR Avg |
|---|---|---|---|
| Qwen1.5-1.8B (SFT) | | 56.6 | 60.7 |
| Qwen1.5-1.8B | Qwen1.5-4B | 62.9 | 66.7 |
| Qwen1.5-1.8B | Qwen2.5-7B | 63.4 | 67.8 |
| Qwen1.5-0.5B (SFT) | | 52.0 | 48.4 |
| Qwen1.5-0.5B | Qwen1.5-1.8B | 54.7 | 51.1 |
| Qwen1.5-0.5B | Qwen1.5-4B | 56.6 | 54.5 |

**Our Distillation Method Generalizes Across Backbones.** Finally, we test our framework's generalizability by replacing the Qwen backbone with the MobileLLaMA family (MLLaMA) in Table 7. Our distillation method remains effective even with this entirely different architecture. This result confirms the robustness and generalizability of our approach, demonstrating that its principles are not tied to a specific model family but are broadly applicable.

Table 7: Experiments with a different LLM backbone.

| Student | Teacher | VQA Avg | CR Avg |
|---|---|---|---|
| MLLaMA-1.7B (SFT) | | 49.7 | 43.7 |
| MLLaMa-1.7B | MLLaMa-3B | 53.1 | 48.9 |

## 7 RELATED WORKS

**Multimodal Large Language Models.** Visual instruction tuning (Liu et al., 2023) has propelled MLLMs to strong performance on diverse benchmarks (Chen et al., 2024; Yang et al., 2024; OpenAI, 2023), yet persistent weaknesses remain in fine-grained visual tasks (Tong et al., 2024; Qi et al., 2025). Early efforts to address these limitations focused on scaling up vision encoders (Lu et al., 2024; Kar et al., 2024) or designing more expressive projectors (Cha et al., 2024; Liu et al., 2024). More recently, the focus has shifted to the internal information flow, with research identifying critical

---

[9]For a detailed explanation of the layer matching strategies, see Appendix F.

[10]For details on the training data, see Appendix G.

failures such as misaligned attention (Jiang et al., 2025; Neo et al., 2025; Darcet et al., 2024) and the dilution of visual features in intermediate layers (Kaduri et al., 2024; Yoon et al., 2025; Chen et al., 2025b). In contrast to these approaches, our analysis focuses on the visual attention dynamics between student and teacher models trained via direct knowledge distillation.

**Knowledge Distillation.** Knowledge distillation (KD) compresses a large teacher model into a smaller, efficient student (Hinton et al., 2015), a technique widely applied to LLMs via logit matching (Sun et al., 2019; Jiao et al., 2020). In the multimodal domain, methods have adapted this for visual grounding (Cai et al., 2024; Feng et al., 2025) or used modular strategies to overcome architectural limits (Shu et al., 2024). The scope has since expanded to distillation across model families (Lee et al., 2025b;a) and a deeper focus on internal mechanics. For instance, recent alignment-oriented methods like VIRAL (Yoon et al., 2025) regularize intermediate representations to preserve visual semantics, moving beyond simple logit-level supervision. While these studies primarily propose new distillation techniques, our work takes a different approach. We instead provide a detailed analysis to identify the specific bottlenecks that hinder the effectiveness of knowledge distillation in MLLMs.

**Compositional Reasoning Benchmarks.** Compositional reasoning, the ability to understand the interplay of objects and their relations, remains a significant hurdle for vision–language models. Initial benchmarks like Winoground (Thrush et al., 2022) first exposed these weaknesses in early models. Building on this, a new generation of more robust benchmarks has emerged to address evaluation biases. These include SugarCrepe (Hsieh et al., 2023), which uses LLM-generated hard negatives; SADE (Ma et al., 2023), which specifically diagnoses and mitigates the language bias found in generative models through a debiased test set that neutralizes syntactic shortcuts; and BiVLC (Miranda et al., 2024), which focuses on bidirectional retrieval with synthetic negatives.

## 8 CONCLUSION

In this work, we aim to enhance the visual perception abilities of Knowledge Distillation(KD)-based Multimodal Large Language Models (MLLMs), which has been largely overlooked by the previous KD studies. To this end, we conduct a systematic analysis and identify visual attention misalignment as a key factor hindering the effective distillation of visual perception from teacher to student. Building on this analysis, we propose CompoDistill, a novel KD framework that incorporates a Visual ATtention alignment (VAT) module to explicitly address this misalignment. Furthermore, we introduce the Teacher Adapter Fetch (TAF) module to ensure that teacher-imposed attention mechanism is compatible with the student's feature space, making synergy with VAT module. Through extensive experiments on VQA and CR benchmarks, we demonstrate that CompoDistill significantly enhances visual perception abilities while preserving strong visual recognition abilities, as achieved in existing KD works. Regarding the limitation and future work, please refer to Appendix P.

We believe that this work provides a novel perspective on the student's visual attention misalignment and makes a contribution to the pursuit of efficient MLLMs, especially in KD-based research, by establishing the first dedicated direction toward enhancing visual perception abilities.

## ACKNOWLEDGEMENT

This work was supported by the Institute of Information & Communications Technology Planning & Evaluation(IITP) grant funded by the Korea government(MSIT) (RS-2025-02304967, AI Star Fellowship(KAIST)) as well as another (IITP) grant funded by the Korea government(MSIT) (RS-2022-II220077, Reasoning, and Inference from Heterogeneous Data)

## ETHICS STATEMENT

Our research contributes to the development of more efficient and accessible Multimodal Large Language Models (MLLMs). By enabling the creation of smaller, yet highly capable student models, our work helps lower the significant computational barriers to deploying advanced AI, promoting its broader adoption. In compliance with the ICLR Code of Ethics, and to the best of our knowledge, we have not faced any ethical issues during this research. Additionally, all datasets and baselines used in our experiments are freely accessible to the public.

## REPRODUCIBILITY STATEMENT

To ensure reproducibility of experiment results throughout the paper, we describe the details of experimental setting and training details in 5.1 and Appendix E, respectively. Furthermore, we provide a source code in https://github.com/ptkjw1997/CompoDistill.

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

# Supplementary Material

*- CompoDistill: Attention Distillation for
Compositional Reasoning in Multimodal LLMs -*

## A    ADDITIONAL VISUALIZATIONS ON ATTENTION MISALIGNMENT

In this section, we provide additional qualitative examples of the **attention misalignment** between the student and the teacher model, a concept introduced in Section 1. To facilitate comprehension, we have highlighted crucial textual components in red, specifically phrases that require relational understanding or accurate attribute recognition. For a comprehensive comparison, we present results for an additional student model (LLaVADI-2B) and our method (CompoDistill), alongside the baseline LLaVA-KD-2B and LLaVA-4B (Teacher). Notably, the visualizations show that CompoDistill produces an attention map remarkably similar to the teacher's, correctly focusing on the key visual areas relevant to the text. These comparisons are illustrated in Figure 6 and Figure 7.

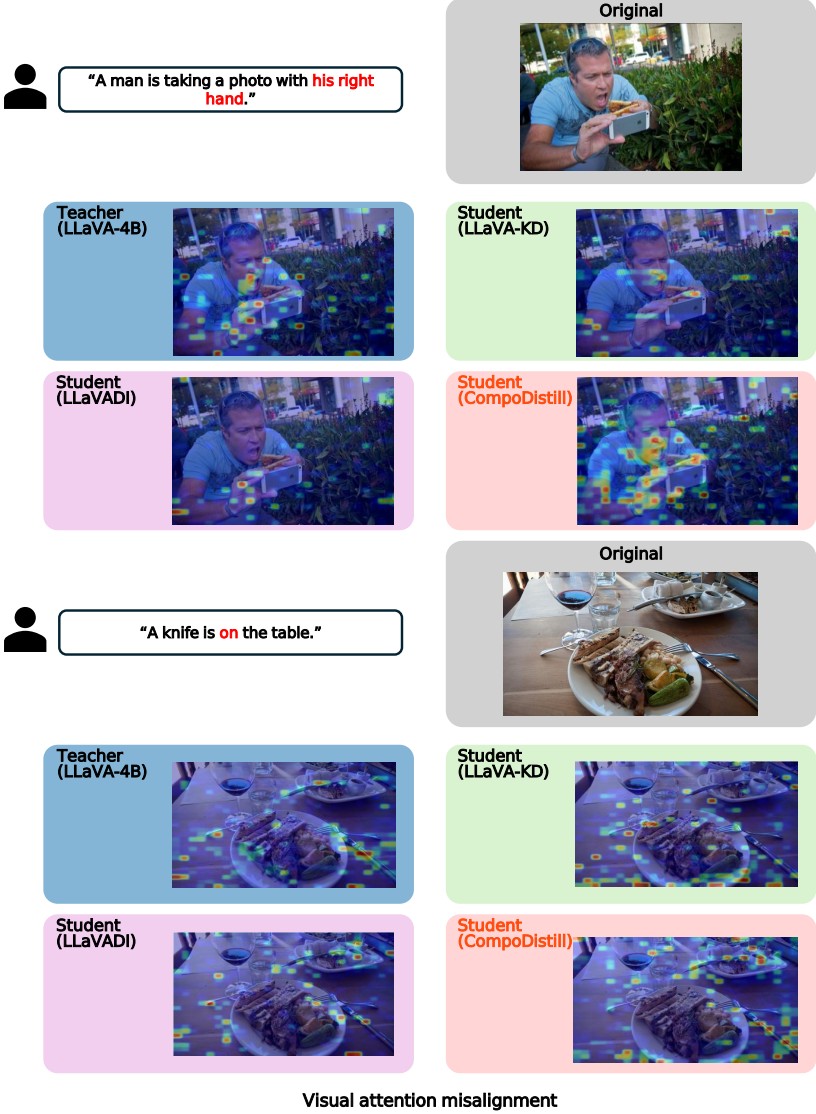

Figure 6: Examples of attention misalignment.

## B    DIFFERENCES BETWEEN VISUAL RECOGNITION AND PERCEPTION ABILITIES

This section provides a detailed explanation of the distinction between visual recognition and visual perception abilities, which are evaluated using Visual Question Answering (VQA) and Composi-

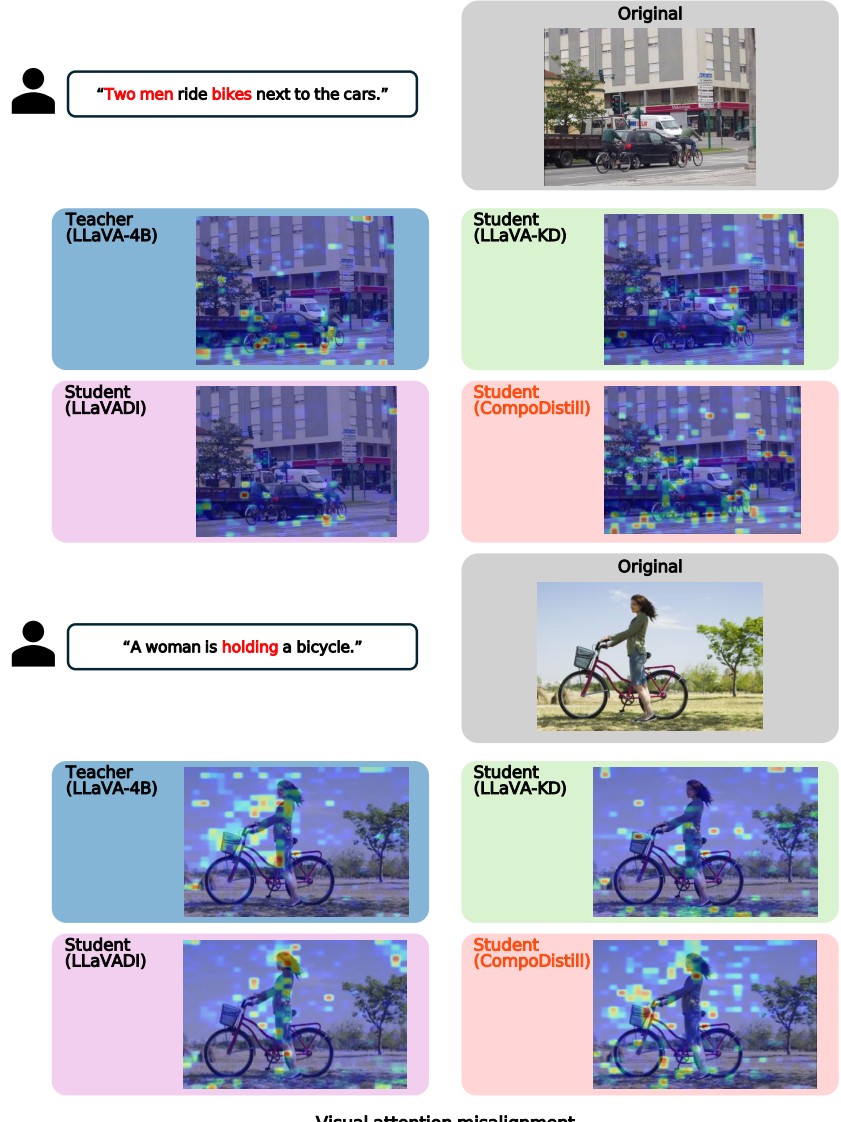

Figure 7: Examples of attention misalignment.

tional Reasoning (CR) datasets, respectively. Also in Figure 8, we illustrated an example of VQA and CR question.

## B.1 VISUAL RECOGNITION

Visual recognition is the foundational ability of a model to **identify and categorize objects, scenes, and their basic attributes within an image.** It fundamentally answers the question, such as "What is in this image?: This process relies on matching learned visual patterns—such as textures, shapes, and colors—to specific labels or concepts. For example, when a model identifies a four-legged furry animal as a "dog", it is performing visual recognition. This ability is analogous to building a vocabulary of the visual world.

Most standard VQA datasets (e.g., VQAv2 (Goyal et al., 2017), Vizwiz (Bigham et al., 2010), and GQA (Hudson & Manning, 2019)) are primarily designed to evaluate this recognition capability. The questions in these datasets are typically direct and fact-based, probing for the presence, count, or simple properties of objects. They can often be answered correctly if the model successfully

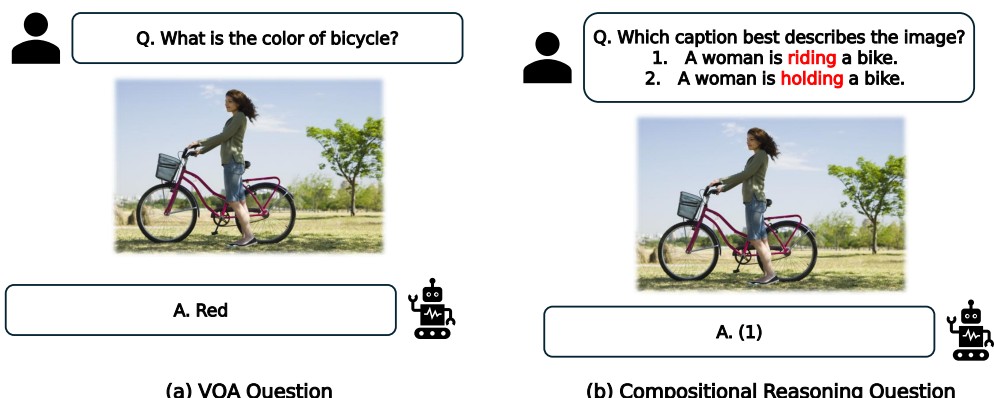

Figure 8: An example of a standard VQA question, which requires simple object identification, versus a Compositional Reasoning (CR) question, which requires accurately distinguishing between two detailed and confusable multiple choices.

recognizes the key objects and their most salient features, without needing to understand complex inter-object relationships.

## B.2 VISUAL PERCEPTION

Visual perception is a more advanced cognitive ability that goes beyond simple identification. It involves **interpreting and understanding the relationships between objects, their precise attributes, their spatial arrangement, and the overall context of the scene**. If recognition is about what, perception is about how and why such as how objects are arranged, how they interact, and why the scene is composed in a particular way. This requires the model to not just list the contents of an image, but to build a coherent, structured understanding of it.

Compositional Reasoning (CR) datasets are specifically designed to evaluate this perceptual ability. The questions are structured to be challenging for models that rely solely on simple keyword matching or recognition. To answer correctly, a model must accurately bind specific attributes to their corresponding objects and correctly interpret the spatial or semantic relationships between them. These questions often test a model's robustness where the correct objects and attributes are present but not in the configuration described by the question.

## C ADDITIONAL EXPERIMENTS ON ATTENTION SIMILARITY

To further validate our findings in Section 3.1, we extend our attention similarity analysis to two additional benchmark datasets: VQAv2 (Goyal et al., 2017) for VQA and Winoground (Thrush et al., 2022) for CR, replicating the experimental setup used for GQA (Hudson & Manning, 2019) and SugarCrepe (Hsieh et al., 2023).

The results, visualized in Figure 9, demonstrate a consistent trend with the conclusions drawn in our main analysis. The analysis reaffirms the critical role of attention over visual tokens in performance improvement. Specifically, on the Winoground (Thrush et al., 2022) (CR) task, where the student model's performance gain over the SFT baseline was marginal, we again observe no significant increase in teacher-student attention similarity. This result corroborates our main argument that higher attention similarity over visual tokens in the visual understanding layers is a key factor for the effective distillation of visual perception.

## D BASELINE METHODS

We include several baselines from two main groups. The first group consists of Knowledge Distillation-based methods, including LLaVADI-2B (Xu et al., 2024), LLaVA-KD-2B (Cai et al., 2024), and LLaVA-MoD-2B (Shu et al., 2024). **To ensure fair comparisons, all KD models were dis-**

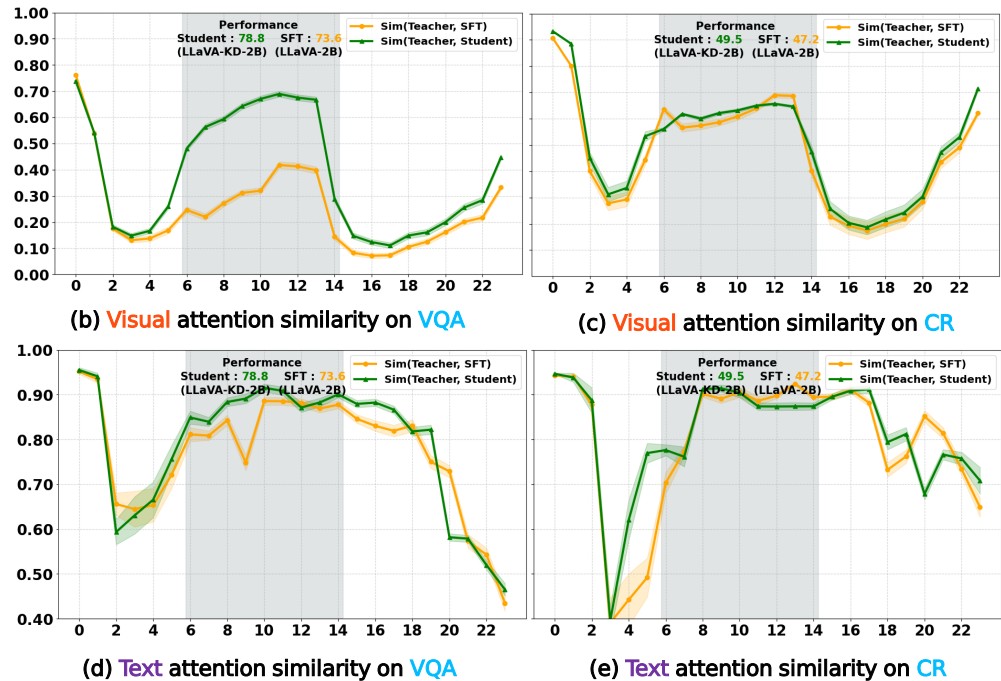

Figure 9: Layerwise attention similarity of visual tokens and text tokens between student/SFT models and the teacher model.

tilled from the same teacher model (i.e., LLaVA-4B) and share the same LLM backbone (i.e., Qwen 1.5). The second group comprises a broad range of General MLLMs with parameter sizes in the range of 1.3B–3B, comparable to that of the compared KD methods, for direct performance comparison. This includes models such as Imp-2B (Shao et al., 2025), Bunny-2B (He et al., 2024), MoE-LLaVA-2B (Lin et al., 2024a), LLaVA-2B (Liu et al., 2024), and Deepseek-VL-1.3B (Lu et al., 2024), MobileVLM-1.7B (Chu et al., 2024), as well as larger models (4B-7B) like MiniCPM-V-2.4B (Hu et al., 2024), CogVLM-7B (Wang et al., 2024) and Qwen2.5-VL-7B (Bai et al., 2025) to provide state-of-the-art context.

# E    ADDITIONAL IMPLEMENTATION DETAILS

## E.1    TRAINING STRATEGY AND HYPER-PARAMETERS

Our model is initialized with a SigLIP-B/14@384 vision encoder and a Qwen1.5-1.8B LLM. For adapter, we use 3-layer MLP. Across all training stages, we use a consistent setup: Each stage is trained for one epoch using AdamW optimizer (Loshchilov & Hutter, 2017). The AdamW optimizer with $\beta_1 = 0.9$ and $\beta_2 = 0.98$, a cosine decay learning rate scheduler, a weight decay of 0.0, and a warm-up ratio of 0.03. We process images at a resolution of 384x384, with input sequence lengths of 729 for the vision encoder and 2048 for the LLM. All training is performed with Float16 numerical precision and Zero2 model parallelism with 8 NVIDIA L40S 48GB GPUs. For the DPT stage, we use the LLaVA-1.5-558K (Liu et al., 2024) dataset with a batch size of 256 and a learning rate of 1e-3. For the DFT and SFT stages, we use the LLaVA-mix-665K (Liu et al., 2024) dataset with a batch size of 128 and a learning rate of 1e-4.

Our training is divided into three distinct stages, each lasting for one epoch:

Distilled Pre-Training (DPT): In this initial stage, our primary goal is to align the visual representations with the language embedding space. To achieve this, we freeze both the vision encoder and the LLM, and exclusively optimize the parameters of the adapter. This stage uses a global batch size of 256.

Distilled Fine-Tuning (DFT): The scope of training is expanded in the second stage to distill knowledge more deeply into the language model. The vision encoder remains frozen, but we now co-optimize both the LLM and the adapter. For this stage, the global batch size is reduced to 128.

Supervised Fine-Tuning (SFT): In the final stage, the model, initialized from the DFT checkpoint, is further refined. The training configuration mirrors the DFT stage: the vision encoder is frozen, while the LLM and adapter are co-optimized. This stage also uses a global batch size of 128 to complete the training process.

The detailed training hyper-parameters are show in Table 8.

Table 8: Training hyper-parameters of each stage.

| Configuration | Distilled Pre-Training | Distilled Fine-Tuning | Supervised Fine-Tuning |
|---|---|---|---|
| LLM | ✗ | ✓ | ✓ |
| Vision Encoder | ✗ | ✗ | ✗ |
| Adapter | ✓ | ✓ | ✓ |
| LLM init. | Qwen1.5-1.8B | Qwen1.5-1.8B | From DFT |
| Vision Encoder init. | | SigLIP-B/14@384 | |
| Image resolution | | 384 x 384 | |
| Vision Encoder Sequence length | | 729 | |
| LLM sequence length | | 2048 | |
| Optimizer | | AdamW | |
| Optimizer hyper-parameter | | $\beta_1 = 0.9, \beta_2 = 0.98$ | |
| Learning rate | | 2e-4 | |
| Learning rate scheduler | | Cosine decay | |
| Weight decay | | 0.0 | |
| Training epoch | | 1.0 | |
| Warm-up ratio | | 0.03 | |
| Global batch size | 256 | 128 | 128 |
| Numerical precision | | Float16 | |
| Model parallelism | | Zero2 | |

## F EXPLANATION OF THE BASELINES FOR LAYER MATCHING STRATEGY

In this section, we provide a detailed explanation of the different layer matching strategies evaluated in our main ablation study (Table 3c). Layer matching is a critical component of attention distillation, as it defines the correspondence between teacher and student layers for knowledge transfer. While our proposed method, **Group matching**, proved to be the most effective, we also explored two alternative baseline strategies to thoroughly investigate the design space. We will describe each of these in turn:

First, **Simple matching**, a straightforward approach that uniformly samples teacher layers to match the number of student layers. Second, **Adaptive matching**. Expanding on (Lee et al., 2025b), this method first computes a matrix of Kullback-Leibler (KL) distances between every student target layer and every teacher target layer. Each student layer is then greedily paired with the teacher layer that has the minimum KL distance, under a consecutive constraint. This constraint ensures that a given student layer can only select a teacher layer that comes after the one selected by the previous student layer, thereby maintaining the sequential integrity of the model's architecture.

## G EXPLANATION FOR THE EXTENDED TRAINING DATA

This section provides detailed information on the training data used for the data scaling component of our main scalability experiments. Our baseline training utilizes a base dataset of approximately 1.2M samples, comprising the **LLaVA-1.5-558K** (Liu et al., 2024) for Distilled Pre-Training (DPT) and **LLaVA-mix-665K** (Liu et al., 2024) for the Distilled Fine-Tuning (DFT) and Supervised Fine-Tuning (SFT) stages.

To scale the data in Sec 6, we incorporate an additional 1.2M samples, originally curated for the Dense-to-Sparse Distillation (D2S) in LLaVA-MoD (Shu et al., 2024). As detailed in Table 9, this extended dataset is a diverse mixture covering a wide range of tasks, including General QA, Grounding, Science, Chart & Document understanding, OCR, and Knowledge-based reasoning.

Table 9: Training dataset of scaling data experiment.

| Stage | Task | Dataset |
|---|---|---|
| Distilled Pre-Training (DPT) | Captioning | LLaVA-1.5-558K (Liu et al., 2024) |
| Distilled Fine-Tuning (DFT) | Conversation | LLaVA-mix-665K (Liu et al., 2024) |
| Supervised Fine-Tuning (SFT) | Conversation | LLaVA-mix-665K (Liu et al., 2024) |
| Data Scaling Experiment | Captioning | ShareGPT4V-100K, TextCaps |
| | Conversation | LLaVA-mix-665K (Liu et al., 2024) |
| | General QA | GQA (Hudson & Manning, 2019), VQAv2 (Goyal et al., 2017), OKVQA (Marino et al., 2019) |
| | Grounding | VG (Krishna et al., 2016), RefCoCo (Yu et al., 2016) |
| | Science | AI2D (Kembhavi et al., 2016), ScienceQA (Lu et al., 2022) |
| | Chart & Doc | DVQA (Kafle et al., 2018), ChartQA (Masry et al., 2022), DocQA (Clark & Gardner, 2017) |
| | OCR | OCRVQA (Mishra et al., 2019), SynthDoG-EN (Kim et al., 2022) |
| | Knowledge | A-OKVQA (Schwenk et al., 2022), GeoQA+ (Cao & Xiao, 2022) |

## H   ADDITIONAL EXPERIMENTS RESULTS

### H.1   DETAILED EXPERIMENT RESULTS ON VAT MODULE

#### H.1.1   DETAILED EXPERIMENT RESULTS ON ATTENTION LOSS TYPE

Table 10: Ablation for various attention loss type.

| Attention Loss Type | Attention Matching | Attention Layers | Visual Question Answering | | | | Compositional Reasoning | | | | |
|---|---|---|---|---|---|---|---|---|---|---|---|
| | | | GQA | TextVQA | MME | Avg | Sugarcrepe | SADE | BiVLC | Winoground | Avg |
| *Ablation on attention loss type* | | | | | | | | | | | |
| | *w/o Attention Loss* | | 60.8 | 56.5 | 66.5 | 61.3 | 78.0 | 67.0 | 62.4 | 48.1 | 63.8 |
| MSE. | Group | intermediate | 59.9 | 55.3 | 65.7 | 60.3 | 79.9 | 62.8 | 49.7 | 65.2 | |
| KL. Div. | Group | intermediate | 60.7 | 55.6 | 65.9 | 60.7 | 80.1 | 67.9 | 63.5 | 50.5 | 65.5 |
| Cos. Sim. | Group | intermediate | 62.2 | 56.4 | 70.1 | 62.9 | 82.9 | 69.4 | 63.3 | 51.1 | 66.7 |

We provide a detailed analysis of the impact of different attention loss functions, as summarized in Table 10. Our goal was to determine the most effective way to quantify the difference between student and teacher attention maps for distillation. The results show that while all tested loss functions improve performance on Compositional Reasoning (CR) tasks over the baseline, their effects on Visual Question Answering (VQA) are mixed. Notably, both Mean Squared Error (MSE) and KL Divergence (KL. Div.) slightly degrade VQA performance.

Cosine Similarity (Cos. Sim.) proved to be the most effective loss function, improving performance across both the VQA and CR domains. We believe this is because it is more crucial for the student model to learn the relative importance of different visual patches, rather than replicating the exact absolute values (MSE) or the probability distribution (KL. Div.) of the teacher's attention scores.

#### H.1.2   DETAILED EXPERIMENT RESULTS ON ATTENTION TARGET LAYERS

Table 11: Ablation for attention target layers.

| Attention Loss Type | Attention Matching | Attention Layers | Visual Question Answering | | | | Compositional Reasoning | | | | |
|---|---|---|---|---|---|---|---|---|---|---|---|
| | | | GQA | TextVQA | MME | Avg | Sugarcrepe | SADE | BiVLC | Winoground | Avg |
| *Ablation on target layers for attention distillation* | | | | | | | | | | | |
| Cos. Sim. | Group | early ($\sim$ 30%) | 61.3 | 55.5 | 66.8 | 61.2 | 80.9 | 67.4 | 59.8 | 46.7 | 63.7 |
| Cos. Sim. | Group | later (70% $\sim$) | 61.2 | 55.7 | 68.4 | 61.7 | 81.8 | 66.8 | 59.7 | 50.0 | 64.6 |
| Cos. Sim. | Group | all | 62.1 | 55.8 | 69.1 | 62.4 | 83.1 | 68.6 | 63.7 | 50.4 | 66.6 |
| Cos. Sim. | Group | intermediate | 62.2 | 56.4 | 70.1 | 62.9 | 82.9 | 69.4 | 63.3 | 51.1 | 66.7 |

We investigate which layers are the most effective targets for attention distillation, with detailed results in Table 11. We compared four strategies: distilling from early layers ($\sim$30%), later layers (70%$\sim$), all layers, and our primary approach of targeting the intermediate layers (**visual understanding layers**).

The results clearly indicate that targeting the visual understanding layers yields the best overall performance on both VQA and CR. While distilling from all layers provides strong and sometimes comparable results, this approach is less computationally efficient. The more focused intermediate

strategy ultimately achieves a better trade-off, slightly outperforming the all-layer approach in overall scores without the associated computational overhead. Conversely, targeting only the early or later layers leads to a noticeable drop in performance.

This confirms our analysis that the intermediate layers, which function as the core **visual understanding layers**, are the most critical for visual-semantic integration. Distilling specifically from this block provides the most potent and effective signal for knowledge transfer, a finding consistent with prior research (Kaduri et al., 2024; Neo et al., 2025).

### H.1.3 Detailed Experiment Results on Layers Matching Strategy

Table 12: Ablation for attention matching strategy.

| Attention Loss Type | Attention Matching | Attention Layers | Visual Question Answering | | | | Compositional Reasoning | | | | |
|---|---|---|---|---|---|---|---|---|---|---|---|
| | | | GQA | TextVQA | MME | Avg | Sugarcrepe | SADE | BiVLC | Winoground | Avg |
| *Ablation on layers matching for attention distillation* | | | | | | | | | | | |
| Cos. Sim. | Simple | intermediate | 60.2 | 56.3 | 68.0 | 61.5 | 81.6 | 67.1 | 63.2 | 50.7 | 65.6 |
| Cos. Sim. | Adaptive | intermediate | 61.4 | 55.9 | 68.6 | 62.0 | 82.1 | 68.3 | 62.8 | 49.5 | 65.7 |
| Cos. Sim. | Group | intermediate | 62.2 | 56.4 | 70.1 | 62.9 | 82.9 | 69.4 | 63.3 | 51.1 | 66.7 |

This part details our ablation on different strategies for matching student and teacher layers, with full results in Table 12 and detailed explanation about each strategy is in Section F. We compared our proposed **Group matching** against two strong baselines: **Simple matching** (uniform sampling of teacher layers) and **Adaptive matching** (pairing based on feature distance). The data shows that while both Simple and Adaptive strategies improve performance over a no-distillation baseline, our Group matching consistently achieves the best results across all benchmarks. Notably, our method demonstrates a clear performance advantage over the next best method, Adaptive matching. Furthermore, our approach is more computationally efficient, as it avoid the complex calculations required to dynamically match layers inherent to adaptive strategies. This suggests that grouping layers provides a more stable and robust signal for the student. By averaging the behavior of a block of teacher layers, our method likely smooths out layer-specific idiosyncrasies and transfers a more generalized attention strategy.

## H.2 Detailed Experiment Results on Relational Hallucination

Table 13: Relational Hallucination Experiment.

| Model | R-Bench | | | Reefknot | | |
|---|---|---|---|---|---|---|
| | Precision | Recall | F1 Score | Perception | Cognitive | Total |
| Teacher (LLaVA-4B) | 66.4 | 97.8 | 79.1 | 45.3 | 78.0 | 67.9 |
| Studnet (LLaVA-4B) | 60.5 | 96.3 | 74.3 | 40.1 | 70.8 | 61.3 |
| LLaVA-KD-2B | 63.5 | 96.2 | 76.5 | 41.8 | 68.8 | 60.3 |
| LLaVA-MoD-2B | 62.5 | 97.6 | 76.2 | 42.0 | 73.1 | 63.4 |
| CompoDistill-2B | 65.7 | 97.6 | 78.6 | 43.2 | 77.3 | 67.9 |

As mentioned in the main text, we conducted experiments on relational hallucination benchmarks to demonstrate a further benefit of our proposed method. Table 13 presents the detailed quantitative results of this evaluation on the R-Bench (Wu et al., 2024) and Reefknot (Zheng et al., 2025) benchmarks.

The results substantiate our claim, showing that CompoDistill significantly outperforms other knowledge distillation methods on the R-Bench benchmark and nearly closes the performance gap to the teacher model. More strikingly, on the Reefknot benchmark, our method achieves performance on par with the teacher model, showcasing its ability to handle complex relational challenges. These findings provide strong evidence that enhancing visual perception via CompoDistill effectively mitigates relational hallucinations and boosts performance on complex visual reasoning tasks.

## H.3 (Additional) Detailed Experiments Results on Training Strategy.

In this section, we provide additional experimental results to analyze the effectiveness of our proposed three-stage training strategy (DPT-DFT-SFT). To demonstrate its efficacy, we investigate the specific impact of each individual stage, with detailed results presented in Table 14.

First, the Distilled Pre-Training (DPT) stage shows a significant impact, particularly on Visual Question Answering (VQA) tasks. As seen by comparing configurations with and without DPT (e.g., (1) vs. (3)), replacing standard pre-training with our distilled approach consistently yields substantial improvements in VQA scores. This highlights DPT's role in building a strong visual knowledge from the teacher.

Next, the Distilled Feature-Tuning (DFT) stage is crucial for enhancing Compositional Reasoning (CR) capabilities. The inclusion of DFT leads to the most significant gains in CR performance across all benchmarks. We attribute this improvement to the effective transfer of the teacher's fine-grained attention patterns through our attention distillation process, which is vital for understanding complex object relationships.

Finally, the Supervised Fine-Tuning (SFT) stage is indispensable. The results from the configuration without SFT (4) show a catastrophic failure on VQA tasks, as the model completely fails to follow task instructions. This demonstrates that SFT is an absolutely critical final step for aligning the model's distilled knowledge with the specific formats and demands of downstream tasks.

Table 14: Ablation for training recipe and teacher adapter fetch module. $^\dagger$:Fail to follow instructions (i.e., answer only single word).

| | Training Recipe | Teacher Adapter | Visual Question Answering | | | | Compositional Reasoning | | | | |
|---|---|---|---|---|---|---|---|---|---|---|---|
| | | | GQA | TextVQA | MME | Avg | Sugarcrepe | SADE | BiVLC | Winoground | Avg |
| | | | *Ablation on different training recipe* | | | | | | | | |
| (1) | PT + SFT | ✓ | 57.9 | 50.6 | 61.2 | 56.6 | 73.3 | 63.9 | 58.5 | 47.2 | 60.7 |
| (2) | PT + DFT | ✓ | 59.5 | 54.6 | 64.8 | 59.6 | 79.1 | 67.3 | 61.8 | 50.3 | 64.6 |
| (3) | DPT + SFT | ✓ | 60.4 | 56.3 | 64.4 | 60.4 | 72.6 | 61.6 | 58.3 | 49.9 | 60.6 |
| (4) | DPT + DFT | ✓ | 1.8$^\dagger$ | 28.0$^\dagger$ | 34.9$^\dagger$ | 21.6$^\dagger$ | 70.2 | 60.3 | 57.3 | 50.8 | 59.7 |
| (5) | DPT + SFT + DFT | ✓ | 60.6 | 56.1 | 65.1 | 60.6 | 81.1 | 67.9 | 63.2 | 48.9 | 65.3 |
| (*) | DPT + DFT + SFT | ✓ | 62.2 | 56.4 | 70.1 | 62.9 | 82.9 | 69.4 | 63.3 | 51.1 | 66.7 |

## H.4 DETAILED EXPERIMENTS RESULTS ON SCALABILITY

### H.4.1 DETAILED EXPERIMENTS RESULTS ON DATA SCALING

This section provides a detailed analysis of our experiments on scaling data quality and data quantity, with full results presented in Table 15. For a clear point of comparison, we also include the performance of a baseline model trained only with Supervised Fine-Tuning (SFT) on the same initial data volume.

Our findings clearly demonstrate the two-fold benefit of our approach. First, by using our distillation method, the model's performance significantly improves over the SFT baseline, even with the same amount of data. This confirms that distillation provides a higher-quality training signal. Second, when we use more of this high-quality data (doubling the training samples), the model's performance is further enhanced. These results validate that our distillation approach offers a substantial performance gain, which is then amplified by increasing the quantity of the training data. A detailed explanation of the training dataset configuration is provided in Appendix G.

Table 15: Data scaling experiment.

| Student LLM | Teacher LLM | # Training Samples | Visual Question Answering | | | | Compositional Reasoning | | | | |
|---|---|---|---|---|---|---|---|---|---|---|---|
| | | | GQA | TextVQA | MME | Avg | Sugarcrepe | SADE | BiVLC | Winoground | Avg |
| | | | *Data scaling experiment* | | | | | | | | |
| Qwen1.5-1.8B (SFT) | | 1.2 M | 57.9 | 50.6 | 61.2 | 56.6 | 73.3 | 63.9 | 58.5 | 47.2 | 60.7 |
| Qwen1.5-1.8B | Qwen1.5-4B | 1.2 M | 62.2 | 56.4 | 70.1 | 62.9 | 82.9 | 69.4 | 63.3 | 51.1 | 66.7 |
| Qwen1.5-1.8B | Qwen1.5-4B | 2.4 M | 62.7 | 56.8 | 70.6 | 63.3 | 89.9 | 67.8 | 68.9 | 53.3 | 69.9 |

### H.4.2 DETAILED EXPERIMENTS RESULTS ON STUDENT AND TEACHER LLM SIZE

This section provides a more detailed analysis of the relationship between teacher and student model sizes in our distillation framework, with full results presented in Table 16.

We first examine a 1.8B parameter student model. As the table shows, distilling from a larger 7B teacher yields a stronger student model compared to distilling from a 4B teacher, with improved performance on both VQA and CR tasks. It is worth noting that we selected a Qwen2.5-7B model as the larger teacher. This decision was made because preliminary evaluations showed that the performance of the Qwen1.5-7B model was not significantly higher than that of the Qwen1.5-4B version;

using the more capable Qwen2.5 architecture ensured a more significant and meaningful gap in teacher capacity for this experiment.

This trend is further validated with a smaller, 0.5B parameter student. The results show a clear progression: the 0.5B student first shows a dramatic improvement over its SFT-only baseline when distilled from a 1.8B teacher. Performance increases again, and quite substantially, when the same 0.5B student is distilled from an even larger 4B teacher. This demonstrates that even smaller student models can effectively absorb the enhanced capabilities of higher-capacity teachers.

In summary, these experiments consistently show that a larger, more capable teacher model is a critical factor in producing a stronger student, regardless of the student's own size. A higher-capacity teacher provides a richer and more effective source of knowledge, successfully transferring more powerful reasoning abilities through our distillation process.

Table 16: Student and teacher LLM Size experiment.

| Student LLM | Teacher LLM | # Training Samples | Visual Question Answering | | | | Compositional Reasoning | | | | |
|---|---|---|---|---|---|---|---|---|---|---|---|
| | | | GQA | TextVQA | MME | Avg | Sugarcrepe | SADE | BiVLC | Winoground | Avg |
| *Large teacher experiment* | | | | | | | | | | | |
| Qwen1.5-1.8B (SFT) | | 1.2 M | 57.9 | 50.6 | 61.2 | 56.6 | 73.3 | 63.9 | 58.5 | 47.2 | 60.7 |
| Qwen1.5-1.8B | Qwen1.5-4B | 1.2 M | 62.2 | 56.4 | 70.1 | 62.9 | 82.9 | 69.4 | 63.3 | 51.1 | 66.7 |
| Qwen1.5-1.8B | Qwen2.5-7B | 1.2 M | 62.9 | 57.1 | 69.8 | 63.4 | 84.1 | 70.7 | 63.8 | 52.6 | 67.8 |
| Qwen1.5-0.5B (SFT) | | 1.2 M | 54.9 | 45.5 | 55.5 | 52.0 | 52.9 | 51.4 | 39.5 | 49.8 | 48.4 |
| Qwen1.5-0.5B | Qwen1.5-1.8B | 1.2 M | 57.3 | 48.9 | 58.1 | 54.7 | 57.3 | 52.8 | 44.0 | 50.4 | 51.1 |
| Qwen1.5-0.5B | Qwen1.5-4B | 1.2 M | 60.1 | 50.3 | 59.5 | 56.6 | 59.5 | 59.0 | 49.1 | 50.5 | 54.5 |

### H.4.3 Detailed Experiments Results on different LLM Backbones

To verify the generalizability of our distillation framework, we conducted an experiment using an entirely different architectural backbone, the MobileLLaMA (Chu et al., 2024) family. This section provides a detailed analysis of these results, which are presented in the Table 17.

Specifically, we applied our distillation method to a 1.4B parameter MobileLLaMA as the student model, using a 2.7B parameter MobileLLaMA model as the teacher. We then compared its performance to a baseline 1.4B student trained with only Supervised Fine-Tuning (SFT). The results clearly show that our distillation method remains highly effective on this new architecture. The distilled student model significantly outperforms the SFT-only baseline across both Visual Question Answering (VQA) and Compositional Reasoning (CR) benchmarks.

This successful application demonstrates the robustness and architectural independence of our approach. It confirms that the core principles of our distillation method are not specifically tailored to the Qwen (Qwen et al., 2025) model family but can be broadly applied to improve the performance of different MLLM backbones, validating the generalizability of our findings.

Table 17: Different LLM backbones experiment.

| Student LLM | Teacher LLM | # Training Samples | Visual Question Answering | | | | Compositional Reasoning | | | | |
|---|---|---|---|---|---|---|---|---|---|---|---|
| | | | GQA | TextVQA | MME | Avg | Sugarcrepe | SADE | BiVLC | Winoground | Avg |
| *Different LLM backbone experiment* | | | | | | | | | | | |
| MobileLLaMA-1.4B (SFT) | | 1.2M | 53.3 | 43.8 | 52.2 | 49.7 | 50.1 | 50.0 | 37.5 | 37.5 | 43.7 |
| MobileLLaMA-1.4B | MobileLLaMA-2.7B | 1.2M | 57.4 | 45.2 | 56.8 | 53.1 | 55.6 | 54.2 | 39.8 | 46.5 | 48.9 |

## I Generalization to Other Architecture

To rigorously validate the robustness and generalizability of the proposed CompoDistill framework beyond the LLaVA architecture, we extended our evaluation to the Qwen3-VL family.

**Experimental Setup.** The Qwen3-VL architecture differs fundamentally from the LLaVA design used in our main experiments. It incorporates distinct training schemes and advanced architectural components, such as Native Resolution ViT (NaViT) and DeepStack. We conducted experiments across varying model scales, ranging from 2B to 8B parameters, to assess scalability. For training efficiency in these ablation studies, we utilized a 50% subset of the Cambrian737K dataset, with the input resolution parameter `max_pixel_values` set to 32,768.

Table 18: Different architecture experiment.

| Student LLM | Teacher LLM | Training Method | Visual Question Answering | | | | Compositional Reasoning | | | |
|---|---|---|---|---|---|---|---|---|---|---|
| | | | GQA | TextVQA | MME | Avg | Sugarcrepe | SADE | BiVLC | Avg |
| Qwen3-VL-2B | | SFT | 57.2 | 50.9 | 70.4 | 59.5 | 85.5 | 79.3 | 64.9 | 76.5 |
| Qwen3-VL-4B | | SFT | 59.7 | 56.7 | 80.0 | 65.5 | 91.6 | 84.4 | 67.2 | 81.1 |
| Qwen3-VL-8B | | SFT | 60.0 | 58.2 | 80.9 | 66.4 | 93.0 | 85.5 | 68.1 | 82.2 |
| Qwen3-VL-2B | Qwen3-VL-4B | Naive KD | 58.9 | 55.3 | 72.1 | 62.1 | 86.2 | 79.6 | 65.8 | 77.2 |
| Qwen3-VL-2B | Qwen3-VL-4B | CompoDistill | 59.4 | 58.9 | 73.2 | 63.8 | 91.0 | 81.4 | 67.1 | 79.8 |
| Qwen3-VL-2B | Qwen3-VL-8B | CompoDistill | 59.5 | 59.2 | 73.7 | 64.1 | 91.5 | 81.5 | 67.4 | 80.1 |

**Results and Analysis.** Table 18 presents the performance comparison on Visual Question Answering (VQA) and Critical Reasoning (CR) tasks. Despite the structural differences in the vision encoder and language model, CompoDistill consistently demonstrates strong performance improvements. This confirms that the visual attention misalignment bottleneck is not specific to LLaVA but is a broader phenomenon in MLLMs, and that our multi-stage distillation strategy generalizes effectively to diverse architectures and larger model scales.

## J DISCUSSION ABOUT TRAINING OVERHEAD

In this section, we provide a comprehensive analysis of the computational efficiency of our proposed framework, CompoDistill, compared to existing state-of-the-art knowledge distillation (KD) methods.

**Quantitative Comparison.** Table 19 summarizes the training overhead of our method against other baselines, LLaVA-KD and LLaVA-MoD. Our framework requires significantly fewer training hours compared to the baselines while maintaining a comparable number of training stages. specifically, the total training time of CompoDistill is approximately **170 hours** (consisting of DPT: 28h, DFT: 88h, and SFT: 54h), which is substantially faster than LLaVA-KD ($\sim$320 hours) and LLaVA-MoD ($\sim$960 hours).

Table 19: Comparison of training overhead between our method and existing baselines. Our method achieves the lowest total training time with streamlined loss components.

| Method | Training Stages | Loss Components | Total Training Time (hrs) |
|---|---|---|---|
| LLaVA-KD | 3 stages | 4 losses | $\sim$320 |
| LLaVA-MoD | 4 stages | 3 losses | $\sim$960 |
| CompoDistill | 3 stages | 3 losses | $\sim$170 |

**Computational Complexity Analysis.** The efficiency of our method stems from the design of the distillation objectives. **LLaVA-KD** introduces a significant computational bottleneck by calculating an expensive correlation matrix for all visual tokens between the student and teacher, resulting in a complexity of $\mathcal{O}(N_v^2 \times d)$, where $N_v$ is the number of visual tokens and $d$ is the hidden dimension. Similarly, the extended training time of **LLaVA-MoD** is attributed to the complex routing and computation within its Mixture of Experts (MoE) architecture. In contrast, our VAT attention loss in **CompoDistill** operates with a much lower time complexity of $\mathcal{O}(N_v \times L)$, where $L$ denotes the number of layers. This linear complexity with respect to visual tokens allows for highly efficient training without sacrificing performance.

**Resource-Constrained Settings.** To further address practicality under resource-constrained environments, the memory overhead of our multi-module framework can be effectively mitigated. Since the teacher model is static during distillation, its outputs (logits and attention maps) can be pre-computed and stored offline. This strategy decouples the teacher's memory requirement from the student's training loop, allowing our method to be deployable on GPUs with limited VRAM capacity.

## K  ADDITIONAL EXPERIMENT ON ATTENTION MIXING

In this section, we investigate the causal relationship between the similarity of visual attention to the teacher model and the performance on vision-centric tasks. While our main paper (Figure 4 in Sec 3.3) suggests this correlation, we conducted additional ablation studies to empirically verify that the performance gain primarily originates from the distilled teacher's *visual* attention.

**Experimental Setup.**  We extended the experimental setup from Sec 3.3 by introducing three variations of attention mixing strategies during inference. Keeping all other conditions identical, we compared the following settings:

- **LLaVA-KD:** The baseline method.

- **+ Teacher Visual Attention (Ours):** Incorporating visual attention from the distilled teacher (Qwen1.5-4B based).

- **+ Teacher Text Attention:** Using the teacher's text attention instead of visual attention.

- **+ Other Visual Attention:** Using visual attention from a larger, external MLLM (Qwen1.5-7B) to test if simply using a stronger model suffices.

- **+ Random:** Injecting random attention maps as a control.

- **+ Teacher Attention:** Using both the teacher's visual and text attention.

**Quantitative Results.**  Table 20 presents the accuracy scores across three compositional reasoning tasks: Swap, Replace, and Add.

Table 20: Different architecture experiment.

|  | Method | Swap | Replace | Add |
|---|---|---|---|---|
|  | # Data Num | 912 | 3,846 | 2,754 |
| (a) | LLaVA-KD | 0.6218 | 0.8023 | 0.8435 |
| (b) | + Teacher Visual Attention (Ours) | 0.6419 | 0.8140 | 0.8747 |
| (c) | + Teacher Text Attention | 0.6157 | 0.8045 | 0.8535 |
| (d) | + Other Visual Attention (Qwen1.5-7B) | 0.6354 | 0.8059 | 0.8567 |
| (e) | + Random | 0.2094 | 0.1708 | 0.1959 |
| (f) | + Teacher Attention (Visual + Text) | 0.6360 | 0.8210 | 0.8531 |

**Analysis.**  The results provide clear empirical support for our framework design. First, utilizing the teacher's visual attention (b) consistently outperforms the baseline (a) across all datasets, confirming that visual attention is the key driving factor. This is further corroborated by the comparison between (b) and (c), which reveals that unlike visual attention, text attention does not yield similar improvements for vision-centric tasks. Second, a critical observation arises from the comparison between (b) and (d), demonstrating that alignment is more critical than mere model power. Although the external model (Qwen1.5-7B) is significantly more powerful than our teacher (Qwen1.5-4B), simply mixing its attention fails to achieve optimal performance. This indicates that attention alignment is effective only when the student and teacher feature spaces are intrinsically aligned.

In conclusion, these experiments justify the design of our VAT and TAF modules, confirming that precise alignment with the teacher's visual attention is a requisite for improving compositional reasoning capabilities.

## L  ADDITIONAL EXPERIMENT ON TEACHER ADAPTER FETCH MODULE

In this section, we address the potential concern that reusing the frozen teacher adapter (Teacher Adapter Fetch, TAF) might constrain the student's flexibility or lock it into the teacher's visual-linguistic idiosyncrasies. We compare our approach against training a student-side adapter from scratch to clarify the trade-offs between efficiency and adaptability.

**On the Lock-in Effect.**  We first clarify that the primary goal of our framework is to faithfully distill the teacher's rich visual capabilities into an efficient student model. Since both models share the same SFT and KD data, mimicking the teacher's generation behavior is a deliberate design

choice to maximize capability transfer rather than a limitation. As supported by prior distillation studies, accurate alignment with the teacher often enhances the student's generalization rather than restricting it.

**Comparative Experiment: Student-Side Adapter Training.** To empirically validate the effectiveness of TAF, we conducted a comparative experiment against a Student-Side-Match approach.

- **Setup:** Instead of reusing the teacher's frozen adapter, we initialized a new vision adapter on the student side and trained it to align with the teacher's representation space. This process required an additional training stage using the LLaVA-1.5-558K dataset before proceeding to the DPT, DFT, and SFT stages.
- **Overhead:** We note that this explicit alignment strategy introduced significant computational overhead, requiring approximately **40 additional training hours**.

**Results and Analysis.** Table 21 compares the performance of the Baseline (SFT), the Student-Side-Match method, and our proposed CompoDistill (TAF). As shown in the results, our TAF module consistently outperforms the Student-Side-Match approach. This demonstrates that reusing the teacher's adapter is not only more computationally efficient but also provides a more effective mechanism for aligning the student's visual space with the teacher's, without compromising adaptability to the target tasks.

Table 21: Comparison of training strategies with computational overhead.

| Student MLLM | Method | Teacher | Extra Time | VQA | | | | CR | | | |
|---|---|---|---|---|---|---|---|---|---|---|---|
| | | | | GQA | TextVQA | MME | Avg | Winoground | SugarCrepe | BiVLC | Avg |
| Qwen1.5-1.8B | SFT | - | - | 57.9 | 50.6 | 61.2 | 54.9 | 47.2 | 73.3 | 58.5 | 59.7 |
| Qwen1.5-1.8B | Student-Side-Match | Qwen1.5-4B | +40h | 56.8 | 52.3 | 68.5 | 59.2 | 50.6 | 78.2 | 60.8 | 63.2 |
| Qwen1.5-1.8B | **TAF (Ours)** | Qwen1.5-4B | **+0h** | **62.2** | **56.4** | **70.1** | **61.9** | **51.2** | **82.9** | **63.3** | **65.8** |

## M  ADDITIONAL EXPERIMENT ON VISUAL ATTENSION ALIGNMENT MODULE

In this section, we investigate the impact of updating attention parameters during the distillation process. Specifically, we address the hypothesis that freezing attention blocks in the middle layers while only updating Feed-Forward Networks might help mitigate potential over-mimicry or improve generalization.

**Experimental Validation: Freezing vs. Tuning.** To empirically verify the necessity of attention tuning, we conducted a comparative experiment with two settings:

- **Freeze:** Freezing the parameters of the attention blocks in the middle layers and updating only the FFNs.
- **Tuning (Ours):** Updating both attention blocks and FFNs as proposed in our VAT module.

**Results and Analysis.** Table 22 presents the performance comparison on VQA and Critical Reasoning (CR) tasks. As shown in the results, freezing the attention blocks leads to a consistent performance degradation across all metrics compared to our standard setting. We interpret this as evidence that the self-attention mechanism is critical for learning the visual reasoning process. Restricting the update of attention parameters acts as a bottleneck, preventing the student from effectively learning the spatial relationships and where to look required for complex visual tasks. This confirms that adapting the attention mechanism is essential for successfully distilling the teacher's visual capabilities.

## N  DISCUSSION ON POSSIBILITY ABOUT TEACHER OVERFITTING

Table 22: Impact of Attention Block tuning in the Visual Attention Alignment (VAT) module.

| Student | Teacher | Attention Block Strategy | VQA | | | | CR | | | |
|---|---|---|---|---|---|---|---|---|---|---|
| | | | GQA | TextVQA | MME | Avg | Winoground | SugarCrepe | BiVLC | Avg |
| Qwen1.5-1.8B | Qwen1.5-4B | Freeze | 61.6 | 55.9 | 64.6 | 60.7 | 49.4 | 80.4 | 62.9 | 64.2 |
| Qwen1.5-1.8B | Qwen1.5-4B | **Tuning (Ours)** | **62.2** | **56.4** | **70.1** | **61.9** | **51.2** | **82.9** | **63.3** | **65.8** |

In this section, we address the concern regarding the risk of the student model strictly overfitting to the teacher's visual focus patterns, which might dampen exploration or degrade robustness. We provide both empirical evidence and a discussion on our loss function design to demonstrate that CompoDistill encourages intrinsic visual reasoning rather than brittle mimicry.

**Empirical Analysis of Focus Patterns.**   To verify whether the student merely mimics the teacher or acquires an intrinsic ability to discern where to look, we analyzed the relationship between attention alignment and task performance using the SugarCrepe dataset. We stratified the test samples based on the visual attention similarity between the teacher and student at visual understanding layers.

Table 23 summarizes the performance and data distribution across different similarity intervals. We observe two key trends:

- **Absence of Over-fitting:** The majority of samples (55.5%) are concentrated in the moderate similarity range of $[0.6, 0.7)$, rather than clustering in the highest similarity intervals (e.g., $[0.9, 1.0]$). This distribution indicates that the student does not simply mimic the teacher's attention map but learns a generalized representation.

- **Robust Intrinsic Reasoning:** Notably, CompoDistill maintains strong performance (0.76) even in intervals where attention similarity is relatively low ($[0.5, 0.6)$). This implies that the student has acquired robust visual reasoning capabilities, enabling it to derive correct answers even when its visual focus partially deviates from that of the teacher.

Table 23: Analysis of performance and sample distribution relative to Attention Similarity.

| Similarity Range | [0.5, 0.6) | [0.6, 0.7) | [0.7, 0.8) | [0.8, 0.9) | [0.9, 1.0] |
|---|---|---|---|---|---|
| **Performance** | 0.76 | 0.87 | 0.82 | 0.79 | 0.82 |
| **Data distribution** | 5.5% | 55.5% | 15.0% | 11.6% | 12.4% |

**Design Choice of VAT Loss Function.**   Furthermore, our specific choice of **Cosine Similarity** for the Visual Attention Alignment (VAT) loss is a deliberate design decision to mitigate overfitting. In our preliminary ablation studies, we observed that stricter alignment objectives, such as Mean Squared Error (MSE) or KL Divergence, led to performance degradation in VQA tasks.

We attribute this to the nature of the loss functions: while MSE forces the student to match the absolute magnitude of the teacher's values (encouraging point-to-point mimicry), Cosine Similarity aligns the *direction* of the attention vectors. This approach teaches the student the *relative importance* of visual tokens without enforcing rigid adherence to the teacher's exact distribution, thereby fostering the learning of underlying reasoning patterns while preserving robustness.

## O   ADDITIONAL EXPERIMENT ON VISUAL GROUNDING

In this section, we empirically validate the impact of the Visual Attention Alignment (VAT) module on localization (grounding) capabilities. Given that CompoDistill demonstrates superior performance in fine-grained visual-centric tasks (e.g., Compositional Reasoning and Hallucination mitigation), we hypothesized that the improved visual attention alignment would naturally extend to precise visual grounding.

**Experimental Setup.**   We evaluated the models on the RefCOCO benchmark. To ensure a fair comparison, all models, including the baselines (SFT and LLaVA-KD), utilized Qwen1.5-1.8B as the backbone LLM. Performance is measured using the Accuracy@0.5 (Acc@0.5) metric.

**Results and Analysis.** Table 24 presents the quantitative results. Our proposed method, CompoDistill, significantly outperforms both the SFT baseline and the LLaVA-KD method. Specifically, CompoDistill achieves an accuracy of **57.8%**, representing a substantial improvement over LLaVA-KD (45.4%). This empirical evidence confirms that the fine-grained visual attention alignment provided by VAT not only improves reasoning capabilities but also translates effectively to precise object localization.

Table 24: Performance comparison on the visual grounding task (RefCOCO-val).

| Method | Backbone LLM | RefCOCO-val (Acc@0.5) |
|---|---|---|
| SFT | Qwen1.5-1.8B | 16.2 |
| LLaVA-KD | Qwen1.5-1.8B | 45.4 |
| **CompoDistill (Ours)** | Qwen1.5-1.8B | **57.8** |

# P LIMITATION AND FUTURE WORK

A potential limitation of CompoDistill lies in its inability to capture the distinct information carried by each head within multi-head attention (Zhao et al., 2024; Kang et al., 2025), as it distills only the average of the teacher's visual attention across all heads. This simplification may lead to information loss by overlooking the diverse roles played by individual heads. Moreover, CompoDistill assumes that the teacher and student MLLMs belong to the same LLM series, as consistency in vocabulary is required for logit-based distillation (Equation 4 in the main paper). This constraint poses a challenge when attempting to distill visual knowledge from teachers belonging to different model families.

As future work, we aim to incorporate the head-specific characteristics of the teacher's visual attention into the distillation process, enabling the student to capture more fine-grained and nuanced visual cues.

