# OpenReview forum: "CompoDistill: Attention Distillation for Compositional Reasoning in Multimodal LLMs"
_ICLR.cc/2026/Conference — ICLR 2026 Poster_

### Official Review · Reviewer_RxKW · 2025-10-22

**Soundness:** 3
**Presentation:** 4
**Contribution:** 4
**Rating:** 8
**Confidence:** 3

**Summary:**

This paper identifies visual attention misalignment as the main bottleneck in distilling visual perception ability in MLLMs through extensive attention-based analyses and large-scale experiments. The authors propose a knowledge distillation framework combining Visual Attention Alignment and Teacher Adapter Fetch based on the findings, which achieves significant improvements in compositional reasoning performance while maintaining strong visual recognition ability, and demonstrates robust generalizability and state-of-the-art performance across multiple benchmarks and model architectures.

**Strengths:**

1. The paper conducts extensive and systematic analysis prior to method design, deeply investigating the root cause of poor visual perception distillation through detailed teacher–student attention comparisons across tasks and layers, and by quantifying the relationship between attention similarity and performance.

2. It proposes an innovative and well-structured knowledge distillation framework that combines Visual Attention Alignment and Teacher Adapter Fetch to address both attention alignment and vision–language feature space mismatch issues, with a Group Layer Matching strategy that preserves richer teacher perception knowledge compared to traditional one-to-one matching.

3. The work presents extensive experiments across multiple compositional reasoning benchmarks, showing significant performance gains while maintaining strong visual recognition ability on VQA datasets, and demonstrating robust generalizability and state-of-the-art results across different backbone architectures.

**Weaknesses:**

1. The analysis in this paper is primarily conducted on the LLaVA, and the conclusions have not been systematically validated on other types of MLLMs with different training data compositions, architectural designs, and especially those employing ViT-based vision encoders. This raises uncertainty about whether the identified “visual attention misalignment” bottleneck and the effectiveness of the proposed CompoDistill framework generalize well to broader scenarios.

2. The proposed KD framework incorporates multiple modules and a multi-stage training strategy, which may introduce higher training overhead compared to existing KD methods. However, the paper does not provide a quantitative comparison of this additional cost nor discuss its applicability under resource-constrained settings, limiting a full assessment of the method’s practicality.

**Questions:**

1. How much additional training cost does your multi-module, three-stage approach incur compared to existing KD methods?
2. Have authors validated the conclusions and method on architectures other than LLaVA?

---

> ### Author Response · Authors · 2025-11-21
> **(1/1) Author Rebuttal**
>
> ## **W1(Q2)**
>
> We genuinely appreciate your insightful feedback regarding the generalizability of CompoDistill across different architectures and model scales.
>
> To rigorously address this and demonstrate the robustness of our method, we conducted additional experiments using the **Qwen3-VL[1] family.** This architecture is fundamentally different from the LLaVA design used in our main experiments, as it incorporates distinct training schemes and advanced techniques such as NaViT (Native Resolution ViT) and DeepStack. Regarding the use of ViT-based vision encoders, we kindly note that despite architectural variations, the vast majority of state-of-the-art proprietary MLLMs currently adopt ViT-based encoders as the standard.
>
> We scaled our experiments across model sizes ranging from 2B to 8B parameters. For training efficiency, these runs utilized a 50% subset of the **Cambrian737K[2] dataset** with max_pixel_values set to 32,768. As shown in the table below, CompoDistill consistently demonstrates strong performance on VQA and CR tasks, validating its effectiveness across diverse architectures and larger model sizes. Furthermore, we observed that the Naive KD baseline yields significant gains in VQA but only marginal improvements in CR tasks. This phenomenon aligns with the motivation illustrated in Figure 1a of our main paper. The fact that CompoDistill achieves substantial additional improvements further provides empirical confirmation of the visual attention misalignment bottleneck.
>
> | Student | Teacher | VQA Avg | CR Avg |
> | :--- | :--- | :--- | :--- |
> | Qwen3-VL-2B | SFT | 59.5 | 76.5 |
> | Qwen3-VL-4B | SFT | 65.5 | 81.1 |
> | Qwen3-VL-8B | SFT | 66.4 | 82.2 |
> | Qwen3-VL-2B | Qwen3-VL-4B (Base-KD) | 62.1 (+ 4.3%) | 77.2 (+ 0.9%) |
> | Qwen3-VL-2B | Qwen3-VL-4B (CompoDistill) | 63.8 (+ 7.2%) | 79.8 (+ 4.3%) |
> | Qwen3-VL-2B | Qwen3-VL-8B (CompoDistill) | 64.1 (+ 7.7%) | 80.1 (+ 4.7%) |
>
> We hope these extended experiments satisfactorily address your concerns regarding the applicability of our method to larger models with different architectures.
>
> We sincerely thank you for your constructive feedback, which has allowed us to demonstrate the generalizability of our framework more comprehensively.
>
> You can refer to the updated PDF in Appendix I, Generalization to Other Architecture, where the revisions are highlighted in red.
>
> [1] Qwen3-VL: Sharper Vision, Deeper Thought, Broader Action, Qwen Team, https://qwen.ai/blog?id=qwen3-vl
>
> [2] Cambrian-1: A Fully Open, Vision-Centric Exploration of Multimodal LLMs, Tong et al., NeurIPS’24
>
> ## **W2(Q1)**
>
> We sincerely thank the reviewer for this valuable comment regarding the potential training overhead of our multi-module, multi-stage framework.
>
> To clarify, our method remains comparable or even more efficient than other KD baselines in terms of training stages, loss components, and total time:
>
> **Training stages:** LLaVA-KD: 3-stage; LLaVA-MoD: 4-stage; **Ours**: 3-stage
>
> **Training losses:** LLaVA-KD: 4 losses; LLaVA-MoD: 3 losses; **Ours**: 3 losses
>
> **Training time:** LLaVA-KD: ~320 hours; LLaVA-MoD: ~960 hours; **Ours**: ~170 hours (DPT: 28h, DFT: 88h, SFT: 54h)
>
> Our framework is much faster because prior baselines introduce significant computational bottlenecks.
>
> **LLaVA-KD** requires the calculation of an expensive **correlation matrix** for all visual tokens between the student and teacher ($O(N^2_v \times d)$). **LLaVA-MoD**’s long training time results from its complex **Mixture of Experts (MoE)** architecture.
> In contrast, our **CompoDistill** operates with a much lower time complexity of $O(N_v \times L)$ when calculating VAT attention loss, making it highly efficient without sacrificing accuracy.
>
> Additionally, under resource-constrained settings, the GPU memory load can be further mitigated by precomputing and storing the teacher model’s outputs offline, rather than generating them on-the-fly during student training.
>
> **In summary,** while our framework introduces multiple components, its overall computational cost is significantly lower than other KD baselines and remains practical even in limited-resource environments.
>
> We greatly appreciate the reviewer’s comment, which gave us the opportunity to clarify the efficiency and practicality of our approach.
>
> You can refer to the updated PDF in Appendix J, Discussion about Training Overhead, where the revisions are highlighted in red.

---

### Official Review · Reviewer_euXb · 2025-11-01

**Soundness:** 2
**Presentation:** 3
**Contribution:** 3
**Rating:** 6
**Confidence:** 3

**Summary:**

This work addresses a gap in the effectiveness of knowledge distillation techniques in distilling visual perception techniques from teacher MLLMs to student models. An analysis is conducted to identify the reason for this issue, which is attributed to visual attention misalignment between the teacher & student models. A new knowledge distillation approach is proposed called CompoDistill which aims to address this issue by aligning visual attention patterns across the student and teacher models. Experiments are conducted which show that the proposed method offers improvements for compositional reasoning tasks.

**Strengths:**

1. This work identifies and addresses an important gap with existing knowledge distillation techniques which has been ignored in previous studies.
2. The analysis of attention pattern misalignment provides good conceptual motivation and support for the proposed CompoDistill method.
3. The experimental results cover a good range of VQA and compositional reasoning datasets.
4. Ablation studies provide interesting insights into the importance of different CompoDistill components (VAT & TAF).

**Weaknesses:**

1. Discussion of related work is relegated to the appendix, which seems inappropriate for a full-length main conference paper.
2. The main experiments (Table 1) provide results for CompoDistill using only a single student/teacher model pair based on Qwen 1.5 (1.8B parameters for the student and 4B for the teacher). This makes it difficult to assess how well the method will generalize to different model architectures and larger model sizes. Limited aggregated results are provided for MobileLLaMA in Table 7, but it seems that the teacher model used here was also quite small.
3. CompoDistill does not seem to offer significant improvements over other baselines on VQA. I understand that compositional reasoning requires more finegrained perceptual abilities, but shouldn't greater visual attention alignment between the student and teacher also lead to some benefits for VQA as well?

**Questions:**

1. From the ablation results (Table 3) it seems that VAT is more critical for compositional reasoning while TAF is more important for VQA. Do you have any insights into why this is the case?
2. Have you evaluated the effectiveness of CompoDistill for distillation from larger models?

---

> ### Author Response · Authors · 2025-11-21
> **(1/N) Author Rebuttal**
>
> ## **(W1)**
>
> We appreciate the reviewer's valuable feedback regarding the organization of the paper. We fully agree that the discussion of related work belongs in the main text for a full-length conference paper.
>
> Accordingly, we have moved the Related Work section to the main body of the manuscript. Please refer to Section 7, where the revised content is highlighted in red.
>
> ## **W2(Q2)**
>
> We genuinely appreciate your insightful feedback regarding the generalizability of CompoDistill across different architectures and model scales.
>
> While Table 7 presented some results using different LLM backbone MobileLLaMA, we acknowledge your concern that the teacher model in that setup was relatively small. To rigorously address this and demonstrate the robustness of our method, we conducted additional experiments using the **Qwen3-VL[1] family.** This architecture is fundamentally different from the LLaVA design used in our main experiments, as it incorporates distinct training schemes and advanced techniques such as NaViT (Native Resolution ViT) and DeepStack.
>
> We scaled our experiments across model sizes ranging from 2B to 8B parameters. For training efficiency, these runs utilized a 50% subset of the **Cambrian737K[2] dataset** with max_pixel_values set to 32,768. As shown in the table below, CompoDistill consistently demonstrates strong performance on VQA and CR tasks, validating its effectiveness across diverse architectures and larger model sizes.
>
> | Student | Teacher | VQA Avg | CR Avg |
> | :--- | :--- | :--- | :--- |
> | Qwen3-VL-2B | SFT | 59.5 | 76.5 |
> | Qwen3-VL-4B | SFT | 65.5 | 81.1 |
> | Qwen3-VL-8B | SFT | 66.4 | 82.2 |
> | Qwen3-VL-2B | Qwen3-VL-4B (Base-KD) | 62.1 (+ 4.3%) | 77.2 (+ 0.9%) |
> | Qwen3-VL-2B | Qwen3-VL-4B (CompoDistill) | 63.8 (+ 7.2%) | 79.8 (+ 4.3%) |
> | Qwen3-VL-2B | Qwen3-VL-8B (CompoDistill) | 64.1 (+ 7.7%) | 80.1 (+ 4.7%) |
>
> We hope these extended experiments satisfactorily address your concerns regarding the applicability of our method to larger models with different architectures.
>
> We sincerely thank you for your constructive feedback, which has allowed us to demonstrate the generalizability of our framework more comprehensively.
>
> You can refer to the updated PDF in Appendix I, Generalization to Other Architecture, where the revisions are highlighted in red.
>
> [1] Qwen3-VL: Sharper Vision, Deeper Thought, Broader Action, Qwen Team, https://qwen.ai/blog?id=qwen3-vl
>
> [2] Cambrian-1: A Fully Open, Vision-Centric Exploration of Multimodal LLMs, Tong et al., NeurIPS’24

---

> ### Author Response · Authors · 2025-11-21
> **(N/N) Author Rebuttal**
>
> ## **(W3)**
>
> We sincerely thank the reviewer for the thoughtful comment and for highlighting the relationship between visual attention alignment and performance on VQA.
>
> Compositional reasoning (CR) requires more fine-grained perceptual abilities. However, following [1, 2], the **visual perceptual abilities** essential for CR and the **visual recognition abilities** relevant to VQA are not simply different levels of difficulty within the same ability, but rather **independent competencies** in distinct domains (with recognition generally being easier).
>
> Accordingly, **CompoDistill** achieves substantial improvements in CR by enhancing visual perceptual ability, an aspect that prior KD methods do not effectively capture. On the other hand, for VQA, existing baseline methods are primarily optimized for **visual recognition**, leading to relatively smaller gains from our approach. We believe the above reasons explain why the improvement in visual perceptual ability (CR) does not directly translate into additional performance gains in visual recognition (VQA).
>
> We would like to respectfully emphasize that the main contribution of our work lies in enhancing compositional reasoning capabilities and other vision-centric performance like Relational Hallucination(i.e., Reefknot and RelBench)  while maintaining strong VQA performance compared to other KD baselines.
>
> We once again thank the reviewer for this insightful observation, which has helped us to better articulate the distinct objectives and contributions of our method.
>
> [1] Learning to See Before Seeing: Demystifying LLM Visual Priors from Language Pre-training, han et al, arxiv’25
>
> [2] LENS: Multi-level Evaluation of Multimodal Reasoning with Large Language Models, Yao et al, arxiv’25
>
> ## **(Q1)**
>
> We sincerely thank the reviewer for this important question and for recognizing the distinct effects of the VAT and TAF modules.
>
> The main motivation and architectural design choices in our paper stem directly from our motivation experiments (Figure 1) and the investigation presented in Section 3 (“Why is the visual perception ability not distilled properly?”).
>
> In Figure 1, we observed that other KD methods which only align the visual representation space achieve strong performance on VQA but fail on CR. This finding suggested that visual representation alignment is crucial for VQA (visual recognition), motivating the design of our TAF module, which provides lightweight yet effective feature alignment compared to other KD methods requiring additional losses.
>
> Subsequently, **Section 3** explores why compositional reasoning (CR) ability is not properly distilled even with visual alignment.
>
> **Section 3.1** identifies key factors for successful distillation, highlighting that **visual attention similarity** between the student and teacher is critical.
>
> **Section 3.2** quantitatively demonstrates that higher visual attention similarity directly correlates with a higher probability of predicting the correct answer token.
>
> **Section 3.3** validates this assumption through a toy experiment designed to isolate the effect of attention guidance.
>
> Through this structured analysis, we found that guiding the student’s visual attention is a critical factor for improved compositional reasoning, even when visual alignment is already applied. This observation led directly to the development of our VAT module.
>
> Given this logical flow, the results in our ablation study are natural and consistent with our design philosophy:
>
> **VAT** plays a central role in improving **compositional reasoning (CR)** through visual attention alignment.
>
> **TAF** contributes more strongly to **VQA** by enhancing visual representation alignment.
>
> We greatly appreciate the reviewer’s insightful question, which allows us to further clarify this complementary relationship between the VAT and TAF modules.

---

### Official Review · Reviewer_BxQL · 2025-11-01

**Soundness:** 3
**Presentation:** 3
**Contribution:** 3
**Rating:** 6
**Confidence:** 4

**Summary:**

In the draft, the authors proposed CompoDistill, a new Knowledge Distillation (KD) framework to better transfer the teacher model's visual perception ability to student model. Multiple experiments show that the proposed KD framework could help improve performance on compositional reasoning tasks while maintaining performance on VQA tasks.

**Strengths:**

1. The authors conducted systematic analysis and identified visual attention misalignment is the key issue that prevent the distillation of the teacher MLLM's visual perception abilities to student model. Both the analysis and findings make sense and are technically sound to me.
2. The two proposed components of Visual ATtention alignment (VAT) and Teacher Adapter Fetch (TAF) are both technically sound to me as well. VAT better align the visual attention and TAF better align the following process.
3. The authors conducted extensive experiments to justify the proposed frameworks.
4. Writing is good and easy to follow.

**Weaknesses:**

1. The current approach seems limited to same model family. How to extend the distillation to cross modal family?
2. The proposed three stage training is very complicated compared to the standard single stage approach. More justification might be beneficial to make sure the gain outweigh the cost.
3. For Table 2. it seems that VAT module is hurting on TextVQA tasks (56.4 vs. 56.5). Any idea what might be the issue?

**Questions:**

Please refer to the paper weakness section for more details and provide more justification accordingly.

---

> ### Author Response · Authors · 2025-11-21
> **(1/N) Author Rebuttal**
>
> ## **(W1)**
>
> We sincerely thank the reviewer for raising this important question about the generalization of our distillation framework beyond the same model family.
>
> As stated in the limitation section of our paper, we acknowledge that differences in vocabularies and tokenizers pose a fundamental challenge for extending knowledge distillation across different LLM or MLLM families. We would like to respectfully note that this remains a general limitation shared by most existing KD methods, rather than an issue specific to our approach.
>
> We acknowledge that recent studies [1–3] have explored the cross-tokenizer distillation challenge, which we view as a valuable and complementary research direction largely orthogonal to the focus of our work.
> Although this direction lies beyond the scope of our current paper, we would like to emphasize that our method is fully compatible with these approaches and could be naturally integrated with them in future research.
>
> As partial evidence, we present a case of distillation from **Qwen 2.5 (Teacher) → Qwen 1.5 (Student)**, which achieves noticeable performance improvements even though the two models differ in training strategies and architectural techniques, while sharing the same tokenizer.
> Although this is a limited example, we believe it provides encouraging evidence that our method can remain effective for cross-model distillation under a shared tokenizer.
>
> | Model Setting | VQA | CR |
> | :--- | :--- | :--- |
> | Qwen2.5 7B -> Qwen 1.5 1.8B | 63.4 | 67.8 |
> | Qwen1.5 1.8B SFT | 56.6 | 60.7 |
>
> [1] Universal Cross-Tokenizer Distillation via Approximate Likelihood Matching, Minixhofer et al. NeurIPS’25
>
> [2] Towards Cross-Tokenizer Distillation: the Universal Logit Distillation Loss for LLMs, Boizard et al. TMLR’25
>
> [3] Enhancing Cross-Tokenizer Knowledge Distillation with Contextual Dynamical Mapping, Chen et al, ACL Findings’25
>
> ## **(W2)**
>
> We sincerely thank the reviewer for the valuable comment and for pointing out the complexity of our three-stage training process.
>
> **1) Regarding the comparison with the LLaVA training recipe,**
>
> We would like to respectfully clarify that our three-stage process (Distilled Pre-Training (DPT), Distilled Fine-Tuning (DFT), and Supervised Fine-Tuning (SFT)) includes one additional stage compared to the standard two-stage LLaVA training recipe (Pre-Training and Supervised Fine-Tuning). This extra DFT stage is introduced to enable more effective transfer of the teacher’s visual knowledge.
>
> **2) Regarding the comparison with standard SFT,**
>
> We respectfully acknowledge that knowledge distillation methods inherently demand greater training resources than standard SFT approaches. Nevertheless, we would like to humbly clarify that this is a well-understood trade-off within the community, aimed at achieving enhanced model performance and more efficient knowledge transfer.
>
> **3) Regarding the comparison with other KD baselines,**
>
> We would like to emphasize that our three-stage method remains significantly more efficient in practice. For example, LLaVA-KD requires approximately **320 hours**, LLaVA-MoD around **960 hours**, whereas our complete three-stage process takes only **170 hours** in total (DPT: 28h, DFT: 88h, SFT: 54h).
>
> Our method is much faster because the baselines introduce significant computational bottlenecks.
> **LLaVA-KD** requires the calculation of an expensive correlation matrix for all visual tokens between the student and teacher ($O(N^2_v \times d)$).
> **LLaVA-MoD**’s long training time results from its complex Mixture of Experts (MoE) architecture.
> In contrast, our **CompoDistill** operates with a much lower time complexity of $O(N_v \times L)$ when calculating VAT attention loss, making it highly efficient.
>
> **In summary,** while our method indeed requires more training cost than standard SFT, it is significantly more efficient than other KD-based approaches. Moreover, when considering inference efficiency, our smaller student model achieves performance highly competitive with much larger models, demonstrating an acceptable trade-off between training cost and deployment efficiency.
>
> We once again thank the reviewer for this insightful comment, which helped us clarify the efficiency and motivation behind our 3-stage training design.
>
> You can refer to the updated PDF in Appendix J, Discussion about Training Overhead, where the revisions are highlighted in red.

---

> ### Author Response · Authors · 2025-11-21
> **(N/N) Author Rebuttal**
>
> ## **(W3)**
>
> We sincerely thank the reviewer for the thoughtful question and for raising this point regarding VAT and its potential impact on the TextVQA task.
>
> First, we consider the decrease from 56.5 to 56.4 to be marginal, so it is difficult to conclude that VAT is hurting the TextVQA task.
>
> However, we can offer a hypothesis for why TextVQA does not show a performance gain, in contrast to the improvements observed on GQA (+1.4) and MME (+3.6).
>
> TextVQA is an **OCR-based visual task** that relies heavily on **visual–text feature alignment** as noted in [1,2]. In contrast, GQA and MME are more complex visual recognition tasks. In our framework, the **TAF** module handles visual feature alignment, while the **VAT** module focuses on visual attention alignment.
>
> Therefore, since TAF is already responsible for improving feature alignment, the performance on TextVQA that mainly depends on such alignment is already sufficiently enhanced. We believe this explains why the subsequent introduction of VAT yields a smaller relative performance gain for TextVQA compared to GQA and MME.
>
> We again appreciate the reviewer’s insightful comment, which helped us clarify the relationship between attention and feature alignment in different types of visual tasks.
>
> [1] ALIGNVLM: Bridging Vision and Language Latent Spaces for Multimodal Document Understanding, Masry et al, NeurIPS’25
>
> [2] Seeing the Image: Prioritizing Visual Correlation by Contrastive Alignment, Xiao et al, NeurIPS’24

---

> > ### Comment · Reviewer_BxQL · 2025-11-28
> >
> > Thanks for the rebuttal, I'll stick with my original positive rating.

---

> > > ### Author Response · Authors · 2025-11-30
> > >
> > > We sincerely appreciate your valuable feedback, which has helped us further improve the paper, as well as the time you've dedicated to this process.

---

### Official Review · Reviewer_4Sdb · 2025-11-01

**Soundness:** 3
**Presentation:** 2
**Contribution:** 2
**Rating:** 4
**Confidence:** 3

**Summary:**

The paper introduces CompoDistill, a knowledge-distillation framework for MLLMs aimed at transferring visual perception (compositional reasoning) rather than only visual recognition. The key diagnosis is visual attention misalignment between teacher and student in the visual understanding layers; the method (i) explicitly aligns attention over visual tokens via a Visual ATtention alignment (VAT) loss with a group layer matching scheme, and (ii) bridges teacher–student vision-language space mismatch via a Teacher Adapter Fetch (TAF) that reuses the teacher’s adapter with a light MLP mapper. A three-stage training (distilled pretraining → distilled finetuning with VAT → SFT) is used. Experiments across VQA and compositional reasoning benchmarks show gains on CR while keeping VQA strong; ablations support VAT/TAF design choices.

**Strengths:**

1. Simple, orthogonal mechanisms. VAT aligns visual-token attention using cosine distance with a one-to-many group layer matching that’s more effective than simple or adaptive matching. TAF pragmatically resolves feature-space mismatch by reusing the teacher’s frozen adapter with a light MLP, making attention transfer workable in practice.

2. Consistent CR gains without sacrificing VQA. Across SugarCrepe, SADE, BiVLC, and Winoground, CompoDistill outperforms KD and maintains competitive VQA, showing perception improves while recognition holds steady. The intermediate-layer focus and cosine loss are empirically validated as strong choices.

3. Breadth and robustness signals. Relational hallucination benchmarks (R-Bench, Reefknot) show additional improvements, hinting at better grounding. Results with a stronger backbone indicate the approach is not tied to a single encoder/LLM stack.

**Weaknesses:**

**1.Causal evidence is still light.**

The analyses show correlation between attention similarity and accuracy and a mild improvement from attention mixing, but they stop short of stronger causal tests (e.g., counterfactual training where only non-visual tokens are aligned). For example, on Figure 2c it shows the difference between student or SFT model with teacher is small on CR; similar trivial difference in Figure 4 is also observed. The author should have done more either empirical or theoretical analysis justifying the analysis. Tightening the causal chain would reduce the risk that VAT is a proxy for other regularization effects.

**2. Adapter reuse may constrain flexibility.**

TAF fetches a frozen teacher adapter and adds a small projector, this could lock the student into the teacher’s visual-linguistic idiosyncrasies and hinder adaptation to novel modalities or encoders. A comparison against training a student-side adapter to match the teacher space (without reuse) would clarify trade-offs.

**3.Risk of overfitting to teacher’s focus patterns.**

For hard CR that requires shifting attention relative to the teacher (e.g., ambiguous or adversarial scenes), strict alignment might dampen student exploration and degrade robustness. Evaluating under label noise, adversarial cropping, or counterfactual edits would reveal whether VAT encourages brittle mimicry.

**Questions:**

1. What happens if you freeze only the attention blocks in mid layers during VAT but allow FFNs to adapt. Does that help avoid over-mimicry?

2. Any evidence that VAT improves localization (pointing/grounding) accuracy measured directly?

---

> ### Author Response · Authors · 2025-11-21
> **(1/N) Author Rebuttal**
>
> ## **(W1)**
> Thank you for raising the important concern regarding the **Visual Attention Similarity with the Teacher Model** and **its relationship to vision-centric task performance (accuracy)**.
>
> **1) Regarding causal evidence between attention similarity and accuracy,**
>
> We conducted additional ablation studies based on the setup in Sec. 3.3 (Figure 4) to more clearly verify that the performance gain observed in Figure 4 primarily originates from the teacher’s visual attention.
>
> Specifically, we tested three variations including “Random Attention Mix”, “Teacher Text Attention Mix”,  “Other MLLM Visual Attention Mix” and "Teacher All Attention Mix (Visual + Text)" to empirically demonstrate that **visual attention from the distilled teacher is the key factor**. In these experiments, we kept all other conditions identical and performed inference under the same settings, varying only the attention mixing strategy.
> | | | Swap | Replace | Add |
> | :--- | :--- | :--- | :--- | :--- |
> | | # Data Num | 912 | 3,846 | 2,754 |
> | (a) | LLaVA-KD | 0.6218 | 0.8023 | 0.8435 |
> | (b) | **+ Teacher Visual Attention (Ours)** | **0.6419** | **0.8140** | **0.8747** |
> | (c) | + Teacher Text Attention | 0.6157 | 0.8045 | 0.8535 |
> | (d) | + Other Visual Attention (Qwen1.5-7B) | 0.6354 | 0.8059 | 0.8567 |
> | (e) | + Random | 0.2094 | 0.1708 | 0.1959 |
> | (f)  | + Teacher Attention (Visual + Text) | 0.6360 | 0.8210 | 0.8531 |
>
> Here, we used **Qwen1.5-4B** as the LLM of the teacher MLLM, while the other MLLM employed the more powerful **Qwen1.5-7B**.
>
> The table shows that only **Teacher Visual Attention** (b) consistently outperforms the baseline (a) across all datasets. Comparing (b) and (c) further confirms that visual attention is the crucial factor for vision-centric tasks such as CR, rather than simply any attention from the teacher (e.g., text attention). Moreover, the comparison between (b) and (d) demonstrates that merely using visual attention from a higher-performing model is insufficient; attention alignment is effective only when the visual spaces are already aligned. Lastly, we investigated the effect of mixing both visual and textual attention (f). While this approach also yielded performance improvements over the baseline, it did not provide significant marginal gains compared to using visual attention alone, reinforcing that the visual component is the primary driver of the observed improvements.
>
> These observations more clearly demonstrate **the relationship between visual attention similarity and vision-task performance.** These experiments provided the rationale for the design of our proposed modules, VAT and TAF. We also believe that the strong compositional reasoning performance of our CompoDistill further provides empirical support for this relationship.
>
> **2) Regarding the performance gap shown in Figure 4,**
>
> In **Figure 2c**, we observe that there is almost no difference in visual understanding layers between the attention similarity of (Student - Teacher) and (SFT - Teacher).
> This small gap indicates that existing KD methods have largely failed to distill the teacher’s **visual perception ability** for compositional reasoning (CR) tasks.
>
> In contrast, as shown in **Figure 4**, by explicitly mixing the teacher’s visual attention into the student’s attention maps, we deliberately increased the (Student - Teacher) attention similarity in visual understanding layers to effectively force a higher similarity gap similar to that observed in **Figure 2b**.  As a result, this intervention led to modest but consistent performance gains in CR tasks, demonstrating that improving attention similarity directly contributes to better compositional reasoning capability.
>
> We would like to emphasize that although the improvement may appear modest, it is consistent and achieved through simple attention mixing without any additional training.
>
> We sincerely appreciate the reviewer’s thoughtful feedback for further strengthening our analysis and clarifying the causal relationship between teacher attention and performance.
>
> You can refer to the updated PDF in Appendix K, Additional Experiment on Attention Mixing, where the revisions are highlighted in red.

---

> ### Author Response · Authors · 2025-11-21
> **(2/N) Author Rebuttal**
>
> ## **(W2)**
>
> We sincerely thank the reviewer for the insightful comment regarding the potential constraint of TAF on adaptation to novel modalities or encoders.
>
> **1) On the potential “lock-in” effect:**
>
> Regarding the concern that TAF might lock the student into the teacher’s visual-linguistic idiosyncrasies and hinder adaptation to new modalities or encoders, we would like to respectfully note that this is not necessarily a performance bottleneck. As supported by prior studies [1, 2], mimicking the teacher MLLM’s generation behavior under an appropriate reward or supervision signal often enhances, rather than restricts, the student model’s learning and generalization.
>
> The main contribution of our paper lies in effectively distilling the rich visual capabilities of the teacher MLLM into the student model. Since both the student and teacher undergo SFT and KD using the same dataset, our work does not aim to surpass the teacher’s performance. Rather, it focuses on **faithfully transferring the teacher’s visual ability to a more efficient student model.** While other research explores surpassing the teacher through self-distillation or large-scale data expansion, we would like to emphasize that such directions are complementary to, but beyond, the scope of our current study.
>
> **2) On training a student-side adapter to match the teacher space:**
>
> We greatly appreciate this constructive suggestion. To investigate potential trade-offs and verify the effectiveness of our design, we conducted the comparative experiment you proposed.
>
> **Experimental Setup:** Instead of reusing the teacher’s frozen adapter (TAF), we trained a new, student-side vision adapter to align with the teacher’s representation space using the same LLaVA-1.5-558K dataset. We then utilized this independently trained adapter throughout the DPT, DFT, and SFT stages. The table below compares the performance of the baseline **(SFT)**, the student-side adapter approach **(Student-Side-Match)**, and our proposed CompoDistill **(TAF)**.
>
> | LLM | Adapter Strategy | VQA Avg | CR Avg |
> | :--- | :--- | :--- | :--- |
> | Qwen1.5-1.8B | SFT | 54.9 | 59.7 |
> | Qwen1.5-1.8B | Student-Side-Match | 59.2 | 63.2 |
> | Qwen1.5-1.8B | TAF (Ours) | 61.9 | 65.8 |
>
> As shown in the results, our **TAF** module consistently outperforms the **Student-Side-Match** approach, even though the latter requires an additional stage taking about 40 hours. This empirical evidence reaffirms that reusing the teacher’s adapter does not hinder the student’s ability to adapt to modalities or encoders. Rather, it demonstrates that TAF is an efficient and effective mechanism for aligning the student’s visual space with the teacher’s, without compromising adaptability.
> Your valuable feedback prompted us to conduct a deeper analysis of the TAF module, allowing us to provide more concrete experimental validation of our design.
>
> We sincerely thank you for guiding us to strengthen our paper.
>
> You can refer to the updated PDF in Appendix L, Additional Experiment on Teacher Adapter Fetch Module, where the revisions are highlighted in red.
>
> [1] Unified Reinforcement and Imitation Learning for Vision-Language Models, Lee et al, NeurIPS’25
>
> [2] EAGLE: Speculative Sampling Requires Rethinking Feature Uncertainty, Li et al, ICML’24

---

> ### Author Response · Authors · 2025-11-21
> **(3/N) Author Rebuttal**
>
> ## **(W3)**
>
> We appreciate your insightful concern regarding the risk of the student MLLM overfitting to the teacher’s visual focus patterns.
>
> **1) Regarding Student MLLM’s Focus Pattern,**
>
> To verify that CompoDistill empowers the student with the intrinsic ability to discern **where to look** rather than merely mimicking the teacher, we analyzed the performance gain relative to the degree of attention alignment. We used SugarCrepe datasets. We stratified the test samples based on the **Visual Attention Similarity** between the teacher and the student at visual understanding layers and measured the performance improvement of CompoDistill over the SFT baseline in each interval.
>
> | Similarity | [0.5, 0.6) | [0.6, 0.7) | [0.7, 0.8) | [0.8, 0.9) | [0.9, 1.0] |
> | :--- | :--- | :--- | :--- | :--- | :--- |
> | Performance | 0.76 | 0.87 | 0.82 | 0.79 | 0.82 |
> | Data distribution | 5.5 % | 55.5 % | 15.0 % | 11.6 % | 12.4 % |
>
> As observed in the table, the majority of samples (55.5%) are concentrated in the moderate similarity range of [0.6, 0.7), rather than clustering in the highest similarity intervals. This distribution indicates that the student model does not simply overfit to  the teacher's attention patterns. Furthermore, CompoDistill maintains strong performance in the lower similarity intervals. This demonstrates that the student has acquired robust intrinsic visual reasoning capabilities, allowing it to derive correct answers even when its visual focus diverges from that of the teacher.
>
> **2) Regarding VAT Loss Function,**
>
> We would like to emphasize that our specific choice of **Cosine Similarity** for the VAT loss function was a deliberate design decision to mitigate overfitting. In our ablation studies, we observed that **stricter** alignment losses like MSE or KL Divergence led to significant performance drops in VQA tasks. Conversely, Cosine Similarity yielded the best performance. We believe this is because Cosine Similarity aligns the direction of the attention vectors by teaching the student **the relative importance** of visual tokens rather than forcing it to match the absolute magnitude of the teacher's values. This approach encourages learning the underlying reasoning pattern without enforcing brittle, point-to-point mimicry.
>
> You can refer to the updated PDF in Appendix N, Discussion on Possibility about Teacher Overfitting, where the revisions are highlighted in red.

---

> ### Author Response · Authors · 2025-11-21
> **(N/N) Author Rebuttal**
>
> ## **(Q1)**
>
> Thank you for your insightful suggestion regarding the freezing of attention blocks.
>
> **1) Clarification on Over-Mimicry,**
>
> First, as detailed in our response to W3, we would like to respectfully clarify that CompoDistill does not suffer from over-mimicry. Our analyses demonstrate that the student acquires an intrinsic capability for "where to see" in visual-centric tasks, rather than simply memorizing the teacher's outputs.
>
> **2) Experimental Validation (Freezing Attention Blocks):**
>
> To empirically verify your suggestion, we conducted an experiment where we froze the parameters of the attention blocks in the middle layers, updating only the FFNs. The comparative results are shown below.
>
> | LLM | Attention Block | VQA Avg | CR Avg |
> | :--- | :--- | :--- | :--- |
> | Qwen1.5-1.8B | Freeze | 60.7 | 64.2 |
> | Qwen1.5-1.8B | Tuning (Ours) | 61.9 | 65.8 |
>
> As shown in the table, freezing the attention blocks resulted in a performance degradation across all tasks compared to our standard setting. We interpret this as evidence that the self-attention mechanism is critical for learning the visual process. Freezing these blocks acts as a bottleneck, preventing the student from effectively learning **where to look** to solve complex visual tasks. This confirms that adapting the attention mechanism is essential for successfully distilling the teacher's visual capabilities.
>
> You can refer to the updated PDF in Appendix M, Additional Experiment on Visual ATtension Alignment Module, where the revisions are highlighted in red.
>
> ## **(Q2)**
>
> We sincerely thank the reviewer for raising this important point regarding the impact of VAT on localization (grounding) capabilities.
>
> Since CompoDistill has demonstrated superior performance in fine-grained visual-centric tasks such as Compositional Reasoning (CR) and mitigating Relational Hallucination compared to other KD baselines (e.g., LLaVA-KD, LLaVA-MoD), we hypothesized that this advantage would naturally extend to visual grounding tasks.
>
> To empirically validate this hypothesis, we conducted additional evaluations on the RefCOCO benchmark [1]. For a fair comparison, all models (including baselines) utilized Qwen1.5-1.8B as the backbone LLM.
>
> | RefCoCo-val | SFT | LLaVA-KD | CompoDistill |
> | :--- | :--- | :--- | :--- |
> | Acc@0.5 | 16.2 | 45.4 | 57.8 |
>
> As presented in the table above, CompoDistill outperforms the baselines on the RefCOCO benchmark. This confirms that the enhanced visual attention alignment provided by VAT not only improves visual reasoning but also translates effectively to precise localization capabilities.
>
> You can refer to the updated PDF in Appendix O, Additional Experiment on Visual Grounding, where the revisions are highlighted in red.
>
> [1] Referitgame: Referring to objects in photographs of natural scenes, Kazemzadeh et al., EMNLP’14

---

> > ### Comment · Reviewer_4Sdb · 2025-11-26
> > **Response to Author**
> >
> > We thank the author for providing more analysis on potential concerns. Most of my concerns have been solved, I have raised my score to 6, best of luck in the next stage!

---

> ### Author Response · Authors · 2025-11-27
>
> We are pleased to hear that your concerns have been addressed. Thank you for acknowledging our efforts and raising the score to 6.

---

### Author Response · Authors · 2025-11-30

Dear Reviewers, AC, and SAC,

First, to our Reviewers, we would like to express our sincere gratitude for the productive discussions that have significantly contributed to the further development of our paper.

Next, for the AC and SAC, to help minimize your workload especially under these unprecedented circumstances, we provide a consolidated overview of the rebuttal outcomes. Below, we summarize the score updates, the consensus on our paper’s strengths, and a reviewer-by-reviewer breakdown of how key concerns were addressed during the discussion phase.

**Rebuttal Score Changes**

- Reviewer 1 [4Sdb] : Marginal Below (4) → **Marginal Above (6)**
- Reviewer 2 [BxQL] : Marginal Above (6) → **Marginal Above (6)**
- Reviewer 3 [euXb] : **Marginal Above (6)** → NA
- Reviewer 4 [RxKW] : **Accept (8)** → NA

**Discussion Summary**

Despite the sudden constraints of the discussion phase, we engaged in active with Reviewers 4Sdb and BxQL. While Reviewer 4Sdb was the sole reviewer who initially rated our paper as Marginal Below (4), they actively participated in the discussion. After we rigorously addressed their key concerns regarding causal evidence and adapter flexibility, they acknowledged the validity of our responses and raised their score to Marginal Above (6). Consequently, all reviewers leaned towards acceptance with positive ratings. Reviewer BxQL also maintained their positive rating (6). Although Reviewers euXb and RxKW could not engage in this rebuttal phase, they have consistently expressed strong support for our contributions with positive evaluations.

Below, we summarize the key strengths highlighted by the reviewers, followed by a reviewer-by-reviewer breakdown of the rebuttal details.

**Summary of Strengths**
1. **Systematic Analysis & Motivation** : The reviewers praised our deep investigation into attention misalignment, confirming that it effectively identifies the root cause and addresses a critical gap in existing studies. [Reviewer RxKW, euXb, BxQL]

2. **Sound & Effective Framework** : Well-motivated and supported by our systematic analysis, the proposed CompoDistill (VAT & TAF) was recognized as an innovative, technically sound, and pragmatic solution. [Reviewer RxKW, BxQL, 4Sdb]

3. **Robust Experimental Results** : All reviewers highlighted the extensive evaluation demonstrating significant gains in compositional reasoning and robustness across diverse benchmarks without compromising VQA performance. [Reviewer RxKW, euXb, BxQL, 4Sdb]

---

> ### Author Response · Authors · 2025-11-30
>
> **Reviwer 4Sdb**
> 1. Regarding the *Causal Evidence of Visual Attention Alignment*, we conducted additional ablation studies, confirming that the teacher’s visual attention is the primary driver of performance gains (Appendix K).
>
> 2. Regarding the *Flexibility of Adapter Reuse (TAF)*, we compared TAF against a separately trained student-side adapter as the reviewer suggested, and the results showed that TAF achieves higher performance and efficiency without limiting adaptability (Appendix L).
>
> 3. Regarding the *Risk of Over-Mimicry*, we analyzed the correlation between attention similarity and performance, finding that the student acquires intrinsic reasoning capabilities rather than merely memorizing the teacher's patterns (Appendix N).
>
> 4. Regarding the *Freezing of Attention Blocks*, we found that freezing attention blocks as the reviewer suggested leads to performance degradation, indicating that adapting the self-attention mechanism is essential for learning effective visual processing (Appendix M).
>
> 5. Regarding the *Impact on Localization*, we validated CompoDistill on the RefCOCO benchmark, demonstrating that improved attention alignment effectively translates to better visual grounding accuracy (Appendix O).
>
>
> **Reviewer BxQL**
> 1. Regarding *cross-model distillation*, we demonstrated effectiveness by distilling Qwen-2.5-7B to Qwen-1.5-1.8B, confirming performance gains despite differences in training strategies and architectural techniques.
>
> 2. Regarding *training complexity*, we clarified that our method is significantly faster than comparable KD baselines (e.g., LLaVA-KD, LLaVA-MoD), supporting this with a time complexity analysis of the most computationally intensive components (Appendix J).
>
> 3. Regarding the *marginal fluctuation on TextVQA*, we noted that the difference is negligible and explained that while TAF sufficiently enhances feature alignment critical for OCR tasks, VAT primarily boosts complex visual reasoning, resulting in distinct performance trends across datasets.
>
> **Reviewer euXb**
> 1. Regarding *generalization across architectures and scales*, we conducted additional experiments using the Qwen3-VL family (up to 8B parameters), confirming that CompoDistill remains effective across different model architectures and sizes (Appendix I).
>
> 2. Regarding the *performance trends on VQA*, we clarified that Compositional Reasoning (perception) and VQA (recognition) are distinct competencies, and our method specifically targets perception improvements while maintaining strong recognition capabilities.
>
> 3. Regarding *the distinct roles of VAT and TAF*, we explained that TAF ensures visual feature alignment (critical for VQA), whereas VAT aligns visual attention (critical for CR), confirming their complementary nature as supported by our motivation experiments.
>
>
> **Reviewer RxKW**
> 1.  Regarding *generalization across architectures and scales*, we conducted additional experiments using the Qwen3-VL family (up to 8B parameters), confirming that CompoDistill remains effective across different model architectures and sizes (Appendix I).
>
> 2. Regarding *training complexity*, we clarified that our method is significantly faster than comparable KD baselines (e.g., LLaVA-KD, LLaVA-MoD), supporting this with a time complexity analysis of the most computationally intensive components (Appendix J).
>
>
> We believe that the comprehensive experiments and clarifications provided throughout this rebuttal process have clearly addressed the reviewers' concerns.
>
> We once again thank you for your time and effort.

---

### Meta-Review · Area_Chair_zLFX · 2026-01-07

**Summary:**

The paper presents a sound and effective framework supported by a systematic analysis and strong motivation. The reviewers highlighted the robustness of the experimental results as a key strength of the work.

While the initial ratings for the paper were mixed, the rebuttal phase proved to be very effective. The authors successfully addressed the concerns raised by the reviewers, leading to a shift in all evaluations toward acceptance. Given the consensus on the paper's quality and the authors' ability to resolve outstanding issues, the Area Chair recommends that the paper be accepted.

**Reviewer Concerns:**

-- addressed --： Causal evidence is still light （Reviewer 4Sdb）

**Reviewer Scores:**

A key concern was the analysis's over-reliance on LLaVA, casting doubt on whether the identified bottleneck and proposed framework generalize to other MLLMs. The authors addressed this in rebuttal with additional experiments across more architectures. The AC finds these results convincingly demonstrate generalization. Although the original reviewer could not fully engage post-rebuttal, the AC believes the concern is resolved and would have raised the score.

---

### Decision · Program_Chairs · 2026-01-26

Accept (Poster)